# Odor mixtures of opposing valence unveil inter-glomerular crosstalk in the *Drosophila* antennal lobe

Ahmed A.M. Mohamed[1], Tom Retzke[1], Sudeshna Das Chakraborty[1], Benjamin Fabian[1], Bill S. Hansson[1], Markus Knaden[1] & Silke Sachse[1]

Evaluating odor blends in sensory processing is a crucial step for signal recognition and execution of behavioral decisions. Using behavioral assays and 2-photon imaging, we have characterized the neural and behavioral correlates of mixture perception in the olfactory system of *Drosophila*. Mixtures of odors with opposing valences elicit strong inhibition in certain attractant-responsive input channels. This inhibition correlates with reduced behavioral attraction. We demonstrate that defined subsets of GABAergic interneurons provide the neuronal substrate of this computation at pre- and postsynaptic loci via $GABA_B$- and $GABA_A$ receptors, respectively. Intriguingly, manipulation of single input channels by silencing and optogenetic activation unveils a glomerulus-specific crosstalk between the attractant- and repellent-responsive circuits. This inhibitory interaction biases the behavioral output. Such a form of selective lateral inhibition represents a crucial neuronal mechanism in the processing of conflicting sensory information.

[1] Department of Evolutionary Neuroethology, Max Planck Institute for Chemical Ecology, Hans-Knoell-Str. 8, 07745 Jena, Germany. These authors jointly supervised this work: Markus Knaden, Silke Sachse. Correspondence and requests for materials should be addressed to S.S. (email: ssachse@ice.mpg.de)

An important role of an animal's brain is to encode, integrate, and interpret olfactory information from the surrounding environment in order to translate this sensory input into a relevant behavioral output. However, most, if not all, odors encountered are not single molecular compounds, but rather complex blends that vary in both valence and ratio of their individual components. Mixture processing has been well studied in vertebrates[1,2] and invertebrates[3–5]. However, the origin and the underlying neuronal mechanisms and its correlate to the behavioral output are still unresolved.

The simplicity of the olfactory system of *Drosophila melanogaster* makes it a favorable model to study mixture processing. The vinegar fly detects odors with olfactory receptor neurons (ORNs) housed in olfactory sensilla on the antennae and the maxillary palps[6]. Most of the ~50 ORN types express one (or two) odorant receptors (ORs) together with the co-receptor (Orco)[7]. All ORNs expressing the same OR innervate the same glomerulus in the antennal lobes (ALs)[8,9], where they synapse onto projection neurons (PNs)[10]. Glomeruli are interconnected by local interneurons (LNs) which are mainly GABAergic and synapse onto both ORNs and PNs[11,12]. The multiglomerular innervation of most LNs supports the idea of their role in global inhibition, thereby ensuring gain control[13–16]. Nonetheless, some of the inhibitory LNs were shown to connect defined subsets of glomeruli[13,16] and might contribute to mixture processing.

Mixture interactions are influenced by the composition and concentration of each component within an odor blend[17]. Recent behavioral studies in flies showed that the attraction of a mixture can be predicted by the behavioral responses towards the individual mixture constituents[18,19]. Another study in mice showed that aversive odors can neutralize attractive odors in a blend, or even turn the mixture into a repellent[20]. However, it still remains elusive how the olfactory circuitry accomplishes mixture processing and where the neuronal correlate to the fly's decision is located along the olfactory pathway.

In order to dissect mixture processing in the fly, we used binary blends of odors having opposing valences and established a mixture ratio at which the repellent odor starts to significantly reduce the attraction towards the mixture. We demonstrate that certain glomeruli contribute differentially to the mixture processing through inter-glomerular crosstalk mediated by GABAergic inhibition, which bias the behavioral output.

## Results

**Establishing mixture ratios of odors with opposing valence**. To investigate how odor blends are perceived, processed, and evaluated by the fly's olfactory system, we chose binary mixtures of odors with opposing valences and first determined the ratio of the mixture components at which the repellent odor reduces the behavioral attraction to the mixture. We used the FlyWalk[21], a behavioral bioassay monitoring odor-guided walking behavior of individual flies (Fig. 1a). Presenting attractive odors usually results in upwind movement, while repellent odors reduce the flies' movement[21]. As a starting point, we picked ethyl acetate ($10^{-2}$) as an attractive odor[18], benzaldehyde ($10^{-1}$) as an aversive odor[21,22] and their binary mixture. In this experiment, flies showed the same attraction to the mixture as to ethyl acetate alone (Fig. 1b, c). We next kept the concentration of the aversive odor constant and blended it with a lower concentration ($10^{-3}$) of ethyl acetate. Although ethyl acetate on its own was still highly attractive, the attraction towards the binary mixture was significantly reduced (Fig. 1d, e). Hence, we had identified the concentration at which benzaldehyde starts to reduce the attraction to the mixture. We define, hereinafter, the attractive mixture as MIX(+) and the mixture with reduced attraction as

MIX(-). To verify that the determined mixture ratio was consistent regardless the behavioral paradigm, we employed an additional two-choice bioassays, the T-maze (Fig. 1f). The flies had to choose between the solvent control (mineral oil) and either the single odors or their binary mixtures MIX(+) or MIX(−). In line with our FlyWalk data, MIX(+) was equally attractive as the attractive odor, while the flies showed a significantly reduced attraction to MIX(−) (Fig. 1g).

**Glomeruli activated by the attractant are inhibited by MIX(−)**. Having established the behavioral output, we next asked how the ratio-dependent switch is encoded in the fly's olfactory system. As it has been shown that the internal state can influence odor-guided behavior as well as odor-evoked responses in the AL[23], we kept the internal state of the flies constant among different experiments (see Methods). Ethyl acetate and benzaldehyde evoke activity in, mostly, non-overlapping glomeruli[24,25]. We first focused on the AL output to analyze whether any mixture processing in form of lateral excitation[26] and/or lateral inhibition[12,27,28] was taking place. We expressed GCaMP6s[29] in PNs under control of *GH146-Gal4*, which labels most of the uniglomerular PNs[30,31]. Using two-photon imaging we monitored odor-evoked signals in PNs applying the same odor delivery system and odor concentrations as used in our behavioral experiments (Fig. 2a). We verified the stimuli using a photoionization detector and SPME GC-MS (Supplementary Fig. 1). We annotated and analyzed all glomeruli that we could confidently identify based on their anatomical position using the in vivo 3D AL atlas (i.e., 34 glomeruli in total)[30] (Supplementary Fig. 2). Using ethyl acetate ($10^{-2}$), benzaldehyde ($10^{-1}$) and their mixture (MIX( + )), we observed that the mixture was linearly represented (Fig. 2b, d and Supplementary Fig. 2a). Ethyl acetate evoked the strongest response in glomeruli DM1, DM2, DM3, and DM4, while benzaldehyde induced strong responses in DL1 and DL5, which is in line with previous data[25]. In general, glomeruli DL1 and DL5 are mostly activated by aversive odors, while glomeruli DM1-DM4 mainly respond to attractive odors and belong to a small and special subset of valence-specific glomeruli[22]. Hence, we name this subset of glomeruli henceforth attractant-responsive or repellent-responsive glomeruli. Interestingly when we measured PN responses to the mixture with reduced attraction, i.e., MIX(−), we noticed a strong inhibition in four out of the 34 glomeruli compared to their activity to the single odor component (Fig. 2c, e and Supplementary Fig. 2b). Notably, these inhibited glomeruli are the four most responsive glomeruli to ethyl acetate. To visualize the odor representations we employed a principal component analysis. MIX(+) was located between its individual components, while MIX(−) was clustered with the repellent odor benzaldehyde (Fig. 2f).

We next examined whether the mixture inhibition is concentration- or ratio-dependent. We established a second pair of MIX(+) and MIX(−) in the FlyWalk by reducing the concentrations 10-fold and measured responses in PNs (Fig. 2g±j and Supplementary Fig. 2c, d). In line with our previous results, only the four attractant-responsive glomeruli were inhibited when stimulated with MIX(−), while the other 30 glomeruli showed a linear mixture representation. We conclude that inhibition of these four attractant-responsive glomeruli is dependent on the ratio between the attractive and aversive odor and is correlated with a reduced behavioral attractiveness of the odor mixture.

**Identity of inhibited glomeruli depends on the repellent**. To address whether other binary mixtures of attractive and aversive compounds induce the same kind of mixture interactions, we tested other odor combinations. We wondered whether activation

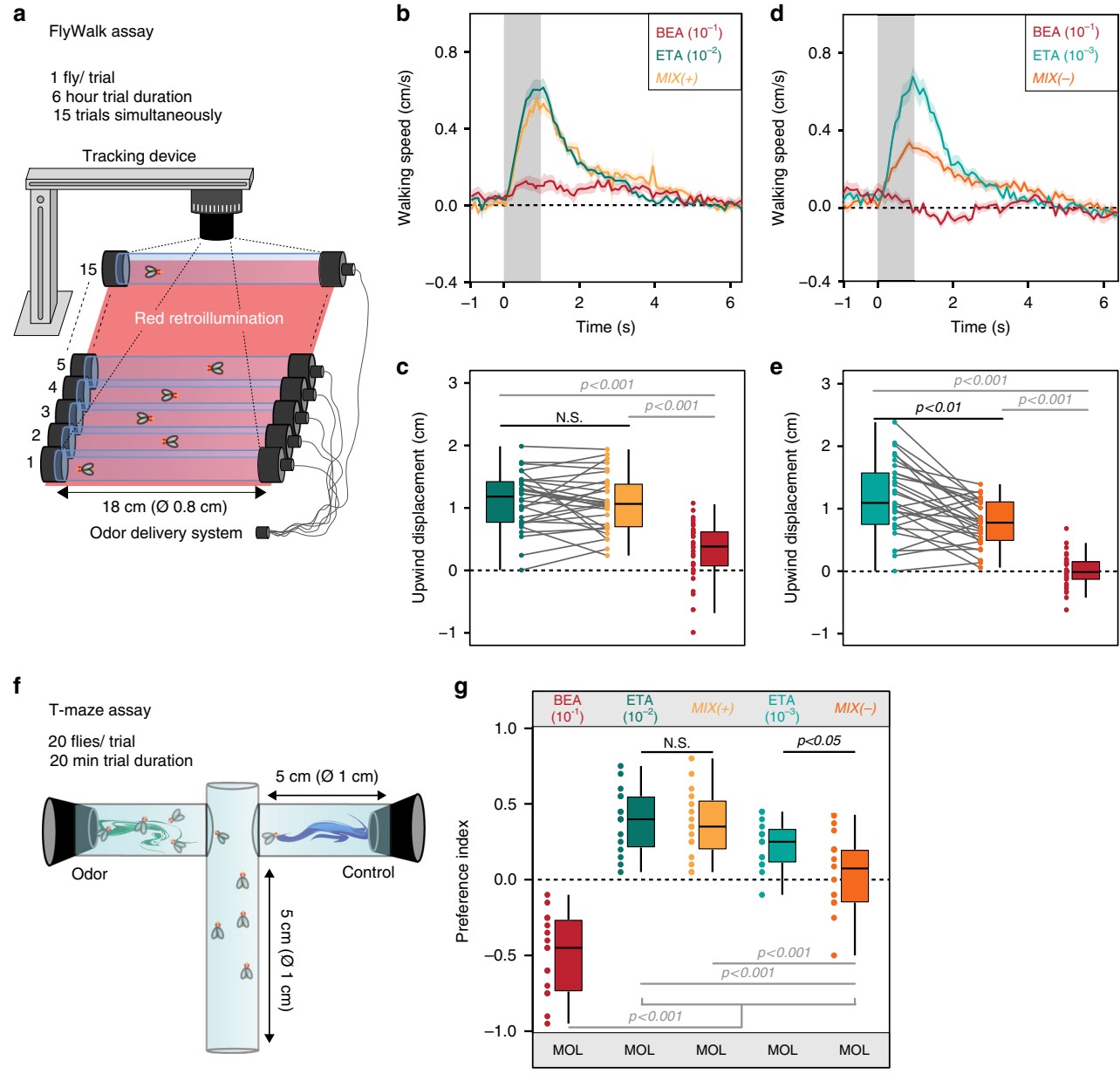

**Fig. 1** Establishing mixture ratio at which repellent odor reduces behavioral attraction. **a** Schematic drawing of the FlyWalk assay. Individual adult female flies are placed in small glass tubes where pulses of single odors or binary mixtures are presented in a continuous airflow (adapted from refs. [33]). **b**, **c** Behavioral responses to ethyl acetate (ETA, $10^{-2}$, blue-green), benzaldehyde (BEA, $10^{-1}$, red), and their binary mixture (MIX( + ), yellow). **b** Quantified behavior from individual flies ($n = 30$) stimulated with ethyl acetate, benzaldehyde, and their binary mixture. Line represents mean upwind speed; shadow indicates SEM. Gray bars in **b**, **d** represent odor pulse (1 s). **c** Box plots represent net upwind displacement of 4 s from odor onset. Colored dots and gray lines represent individual flies (Wilcoxon signed rank test). **d**, **e** Same as in **b**, **c** but for a lower concentration of ethyl acetate ($10^{-3}$). **d** Quantified behavior from individual flies ($n = 30$) stimulated with ethyl acetate (bright blue-green), benzaldehyde (red), and their binary mixture (dark orange). **e** Box plots represent net upwind displacement of 4 s from odor onset. **f** Schematic drawing of the T-maze assay. **g** Box plots showing behavioral preference indices in the T-maze assay to benzaldehyde ($10^{-1}$), ethyl acetate ($10^{-2}/10^{-3}$), MIX( + ) and MIX( − ) against the solvent control (MOL) ($n = 16$–17; one-way ANOVA with posthoc Tukey test). Box plots here and in all following figures represent the median value (horizontal line inside the box), the interquartile range (height of the box), and the minimum and maximum value (whiskers, excluding the outliers) of each experimental group

of solely one of the two repellent-responsive glomeruli is sufficient to induce mixture interactions. To investigate this we selected the odor methyl salicylate, which is a *Drosophila* repellent[18,22] and activates only glomerulus DL1 at the used concentration of $10^{-3}$ (Supplementary Fig. 3a, b)[24]. We blended methyl salicylate with different concentrations of ethyl acetate and determined another set of MIX( + ) and MIX( − ). In this odor combination, behavioral attraction towards high concentration of

ethyl acetate ($10^{-2}$) was not affected, while the attraction of the lower concentrated attractant ($10^{-3}$) was significantly reduced by the repellent in the mixture (Fig. 3a, b). When testing the different mixtures in calcium imaging experiments, we found inhibition in only two out of the four attractant-responsive glomeruli (DM1 and DM4) during stimulation with MIX( − ), while MIX( + ) was linearly represented (Fig. 3a, b and Supplementary Fig. 3a, b). This implies that activation of glomerulus DL1 might

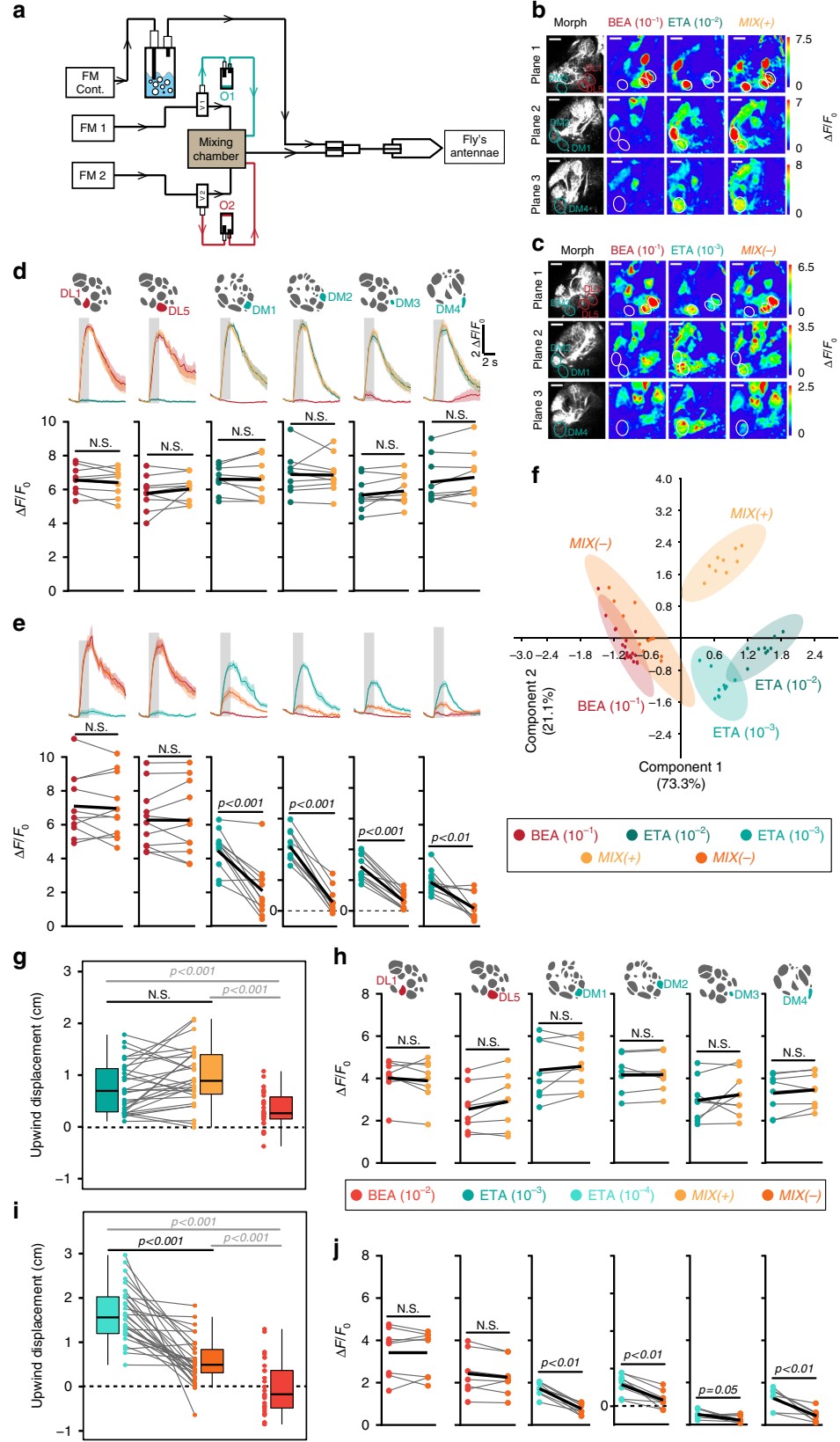

induce inhibition of DM1 and DM4, while activation of DL5 might be required for inhibition of DM2 and DM3 – an assumption that we pursue in more detail in the next section.

We then asked whether the mixture-induced inhibition occurs also in other glomeruli that are activated by attractive odor compounds. We therefore chose balsamic vinegar as it is one of the most attractive odors for vinegar flies[32]. Moreover, as it contains ethyl acetate, it activates overlapping but also additional glomeruli (Supplementary Fig. 3c). Again, we determined the concentrations at which the attraction to balsamic vinegar was

**Fig. 2** Glomeruli responding to the attractive odor reveal mixture inhibition. **a** Schematic of odor delivery system connected to 2-photon microscope. FM flowmeter, cont. continuous, O1/O2 odor 1/odor 2. **b, c** Representative odor-evoked calcium responses in PNs from three focal planes. Gray-scale images represent AL structure with identified glomeruli. Calcium responses are shown to ethyl acetate ($10^{-2}/10^{-3}$), benzaldehyde ($10^{-1}$), MIX ($+$) and MIX($-$). Scale bar $= 20\,\mu M$. **d** Mean PN activity of strongest activated repellent-responsive (DL1, DL5, red) and attractant-responsive glomeruli (DM1, DM2, DM3, DM4, blue-green) during stimulation with ethyl acetate ($10^{-2}$, blue-green), benzaldehyde ($10^{-1}$, red) and their binary mixture (MIX($+$), yellow). Odor responses of all annotated glomeruli (in total 34) are shown in Supplementary Fig. 2. Upper panel, averaged time traces of calcium signals with SEM (shadow); gray bar represents 2 s odor stimulation. Lower panel, mean fluorescence signals during odor stimulation; individual flies are given by single dots and lines; mean is indicated by black thick line ($n = 9$, paired $t$-test). Pairwise comparisons of mixture responses to the response of the strongest single component (i.e. either ethyl acetate or benzaldehyde) are shown for each animal. **e** Same as in **d** for ethyl acetate at $10^{-3}$ (bright blue-green), benzaldehyde ($10^{-1}$, red) and MIX($-$) (orange) ($n = 11$, paired $t$-test). **f** PCA of six most activated glomeruli during stimulation with the odors shown in **b**–**e**. Colored dots represent individual measurements. Shadows represent 95% ellipses for each odor. MIX($-$) and benzaldehyde representations are not significantly different (one-way ANOSIM, Rho similarity index). **g** Box plots represent net upwind displacement in the FlyWalk within 4 s following stimulation with ethyl acetate ($10^{-3}$, bright blue-green), benzaldehyde ($10^{-2}$, bright red), and MIX($+$) (yellow). Colored dots and gray lines represent individual flies. ($n = 30$, Wilcoxon signed rank test). **h** Mean PN activity of strongest activated repellent- and attractant-responsive glomeruli during stimulation with the odors from **g** ($n = 8$, paired $t$-test). **i** Same data as in **g** with ethyl acetate at $10^{-4}$ (turquoise, $n = 30$, Wilcoxon signed rank test). **j** Mean PN activity of strongest activated repellent- and attractant-responsive glomeruli during stimulation with the odors from **i** ($n = 8$, paired $t$-test)

reduced when mixing it with benzaldehyde (Fig. 3c). Using functional imaging of PNs, we again observed an inhibition of the four attractant-responsive glomeruli (DM1, DM2, DM3, and DM4). Interestingly, additional attractant-responsive glomeruli (activated by balsamic vinegar) were not inhibited by benzaldehyde (Fig. 3c and Supplementary Fig. 3c).

Next, we mixed balsamic vinegar with the repellent geosmin, which activates glomerulus DA2[33], and analyzed whether the same mixture interactions would occur. We established a MIX($-$) for this odor combination (Fig. 3d) and monitored the odor representation in PNs. Surprisingly, all activated glomeruli exhibited the same activity to MIX($-$) compared to the individual odorants (Fig. 3d and Supplementary Fig. 3d).

In sum, these findings demonstrate that activation of different ORN types by binary mixtures of odors with opposing valences induces different inhibitions at the PN level in the AL. Notably, we never observed any mixture inhibition of the repellent-responsive glomeruli.

**Specific glomerular crosstalk.** Our results thus far show that glomeruli DL1 and DL5 might distinctively inhibit the four attractant-responsive input channels. We, therefore, postulate a glomerulus-specific crosstalk between the attractant- and repellent-responsive glomeruli. To test this, we first investigated the effect of selectively silencing the input to the repellent-responsive glomeruli at a functional and behavioral level. To do so, we monitored calcium signals from PNs after stimulation with MIX($+$), MIX($-$), and the individual odors benzaldehyde and ethyl acetate in flies where DL1 or DL5 were individually silenced using a mutant background of Or10a or Or7a, respectively (Fig. 4a–f). Both mutants revealed no odor-evoked PN activity in the corresponding glomerulus, indicating that lateral excitation seems not to take place in these cases[34]. We noted that stimulation with MIX($+$) did not result in any inhibition in flies bearing one of the two mutant backgrounds (Supplementary Fig. 4a, b). However, the MIX($-$)-induced inhibition in DM1 and DM4 was abolished in flies with a non-functional DL1 glomerulus, while the inhibition of glomeruli DM2 and DM3 was still visible (Fig. 4b, c). Interestingly, when we silenced DL5, the MIX($-$)-induced inhibition was abolished in DM3 and reduced in DM2, while DM1 and DM4 were unaffected (Fig. 4e, f).

We then wondered whether silencing the repellent-responsive receptors would also affect the behavioral output. We turned to the T-maze that allowed us to monitor strong repulsion in odor-guided behavior (Fig. 4g). Using *w1118* as a control line, we

observed a robust repulsion to benzaldehyde ($10^{-1}$), while both concentrations of ethyl acetate ($10^{-2}$ and $10^{-3}$) were highly attractive (Fig. 4h). Notably, when the input to DL1 was silenced, mutant flies were attracted to MIX($-$), while they were still repelled by benzaldehyde alone. As the inhibition of the attractant-responsive glomeruli DM1 and DM4 was subsequently abolished, the modified behavioral output indicates the importance of these glomeruli for behavioral attraction. Indeed, silencing the input to these two glomeruli by expressing *UAS-Kir2.1*[35], odor attraction towards ethyl acetate as well to both mixtures was abolished and even shifted to aversion for MIX($+$) and MIX($-$) (Supplementary Fig. 4c). This finding is consistent with previous studies showing the significant role of Or42b and Or59b (i.e. DM1 and DM4) for flies' odor attraction[22,36]. Surprisingly, silencing glomerulus DL5 did not change the flies' preference towards any of the tested odors (Fig. 4h). We therefore hypothesize that not all activated glomeruli might be crucial for odor valence coding and that only very few, special glomeruli show valence-specificity – an assumption that needs to be tested further. Notably, Or10a$^{-/-}$ flies showed still a strong repulsion to benzaldehyde suggesting that this avoidance is mediated through several channels in a combinatorial way.

We next asked whether activation of a specific ORN population that responds to aversive odors is sufficient to induce the observed inhibition. To do so, we replaced the aversive odor by optogenetic activation of the repellent-responsive glomeruli, while we simultaneously stimulated the antennae with ethyl acetate. The red-shifted channelrhodopsin CsChrimson[37] was expressed in Or10a- or Or7a-expressing ORNs, respectively, while we monitored calcium signals in PNs (Fig. 5a, d). We first calibrated the light intensity required to evoke activity that simulated the physiological response to odor stimulation (Fig. 5b, e and Supplementary Fig. 5a, b, f, g). Artificial photoactivation of Or10a-expressing ORNs (i.e. DL1) inhibited the calcium responses to ethyl acetate in glomeruli DM1 and DM4 (Fig. 5c, and Supplementary Fig. 5d, e). On the other hand, when we photoactivated DL5 during stimulation with ethyl acetate, the activation of DM3 was significantly inhibited (Fig. 5f and Supplementary Fig. 5i, j). However, excitation of glomerulus DM2 was only slightly reduced by artificial activation of either Or7a- or Or10a-expressing ORNs. Possibly, inhibition of DM2 might require co-activation of both repellent-responsive glomeruli DL1 and DL5 and weakly activated glomeruli. As expected, control flies (i.e. flies fed on artificial food with no all trans-retinal) showed no activation in DL1 or DL5 with light stimulation, and displayed an unmodified activation of the

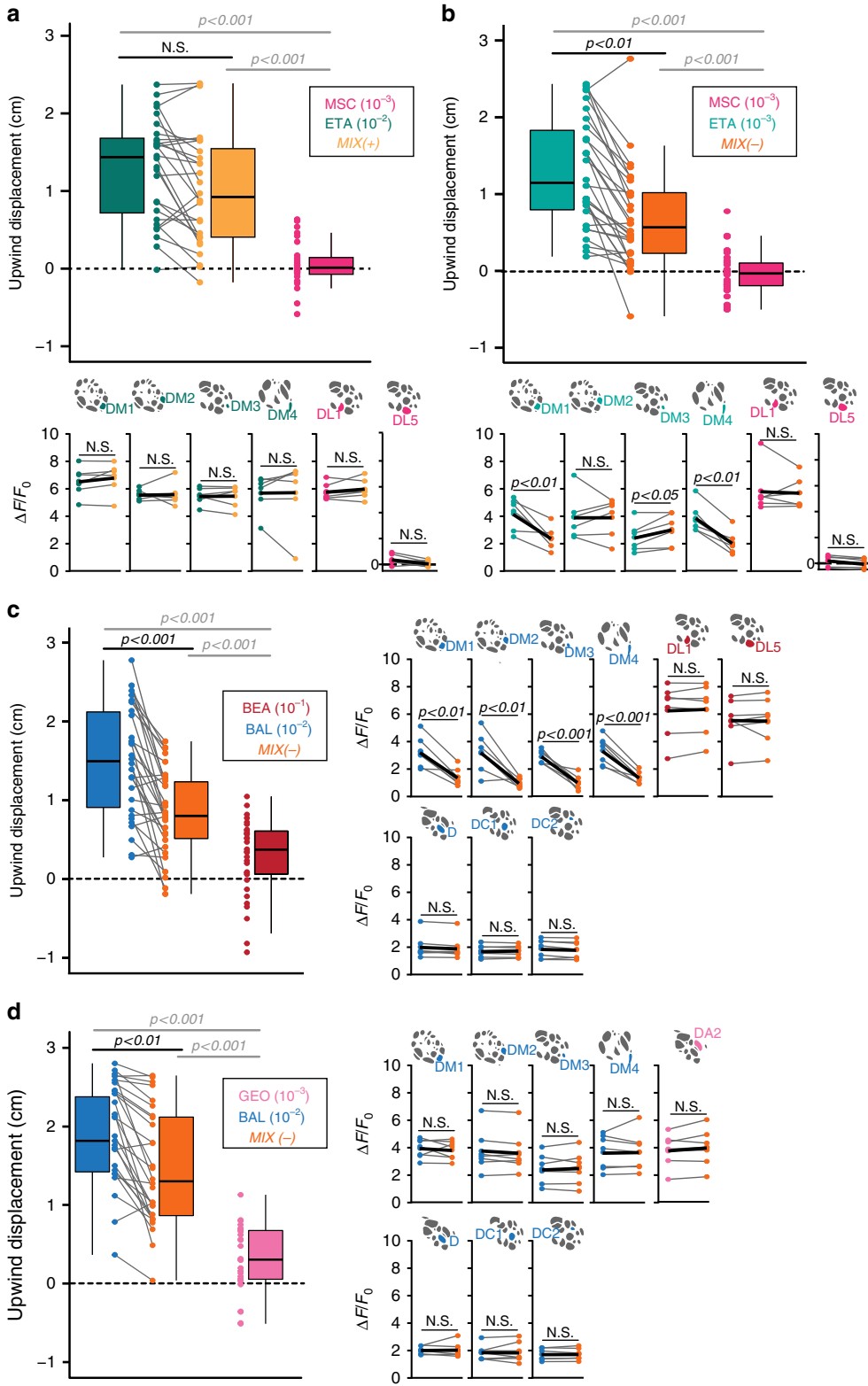

attractant-responsive glomeruli during light and ethyl acetate stimulation (Supplementary Fig. 5c, e, h, j). These results are consistent with the data obtained by silencing single ORN types (Fig. 4).

Next we wondered whether the behavioral response to MIX(−) could be mimicked by replacing the aversive odor with optogenetic activation of the two repellent-responsive glomeruli. CsChrimson was expressed in Or10a- or Or7a-expressing ORNs

and the behavioral response to ethyl acetate was monitored in the T-maze (Fig. 5g). Artificial activation of DL1 and DL5 combined with stimulation of ethyl acetate at $10^{-2}$ was as attractive as ethyl acetate alone which resembles the response to MIX(+). As predicted, a lower concentration of ethyl acetate ($10^{-3}$) combined with artificial activation led to significantly reduced attraction and mimicked the behavioral response to MIX(−). Notably, photo-activation of DL1, DL5, or both resulted in aversive behavior

**Fig. 3** Different binary mixtures evoke glomerulus-specific inhibitions. **a**, **b** Upper panel, box plots represent net upwind displacement in the FlyWalk within 4 s following stimulation with ethyl acetate ($10^{-2}$ and $10^{-3}$, blue-green/bright blue-green), methyl salicylate ($10^{-3}$, magenta) and their binary mixtures (MIX(+) and MIX(−), yellow/orange). Colored dots and gray lines represent individual flies ($n = 30$, Wilcoxon signed rank test). Lower panel, mean PN activity of strongest activated attractant- and repellent-responsive glomeruli during stimulation with the odors from **a** or **b**, respectively ($n = 6$, paired $t$-test). **c** Left, box plots represent net upwind displacement in the FlyWalk within 4 s following stimulation with balsamic vinegar ($10^{-2}$, blue), benzaldehyde ($10^{-1}$, red), and their binary mixture (MIX(−), orange) ($n = 30$, Wilcoxon signed rank test). Right, mean PN activity of strongest activated attractant- and repellent-responsive glomeruli during stimulation with the odors used in the FlyWalk ($n = 6$, paired $t$-test). **d** Left, box plots represent net upwind displacement in the FlyWalk within 4 s following stimulation with balsamic vinegar ($10^{-2}$, blue), geosmin ($10^{-3}$, pink), and their binary mixture (MIX(−), orange; $n = 30$, Wilcoxon signed rank test). Right, mean PN activity of strongest activated attractant- and repellent-responsive glomeruli during stimulation with the odors used in the FlyWalk ($n = 6$, paired $t$-test). Odor responses to the different mixture combinations of all annotated glomeruli (in total 34) are shown in Supplementary Fig. 3

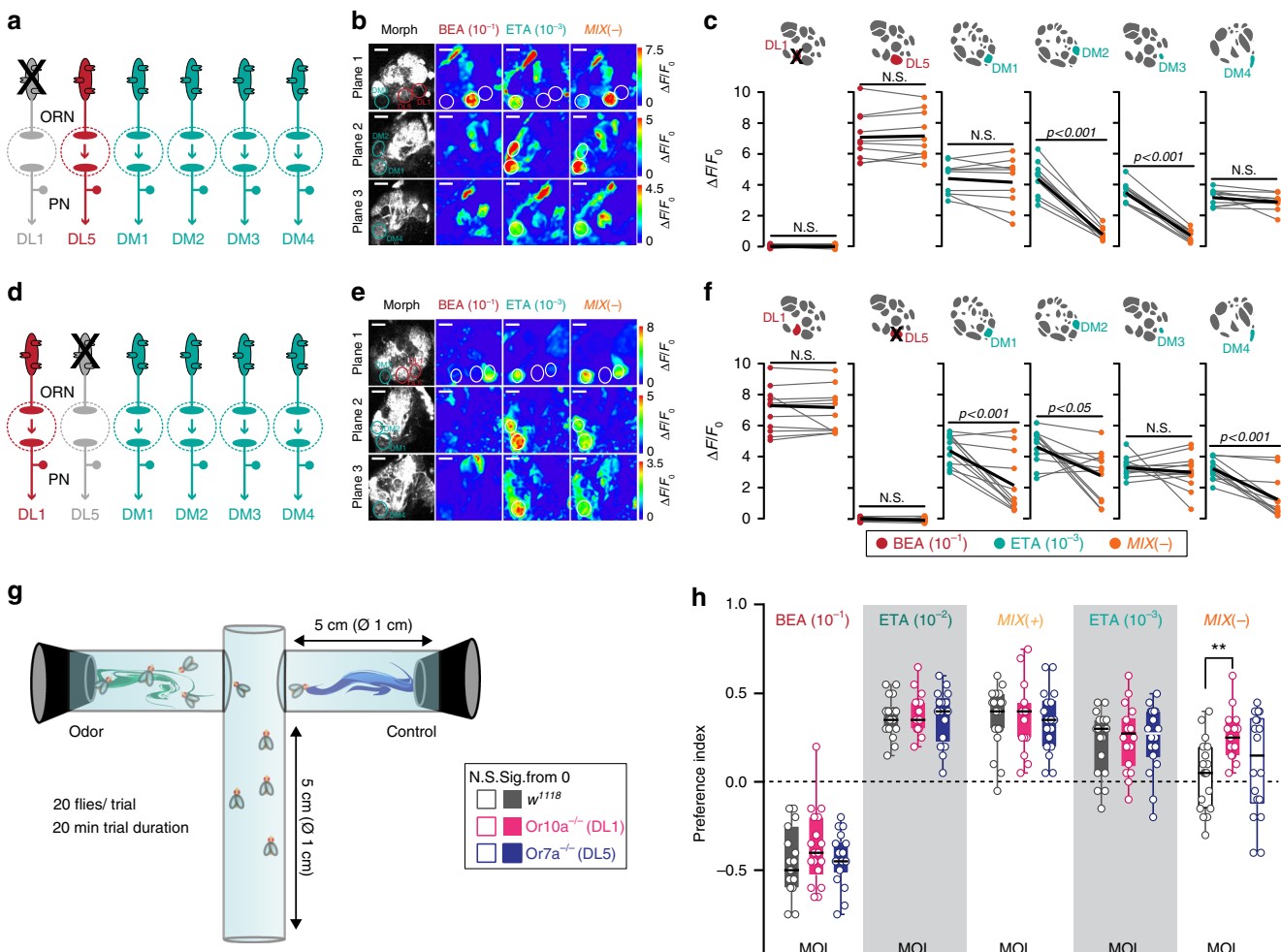

**Fig. 4** Selective silencing of input channels reveals glomerulus-specific inhibition. **a** Schematic of the experimental design: Or10a-expressing ORNs (targeting DL1) are not functional in a Or10a$^{-/-}$ mutant background. Color code indicates glomerulus-specific activation by the attractive (bright blue-green) or repellent (red) odor. **b** Representative odor-evoked calcium responses in PNs from three focal planes of Or10a$^{-/-}$ mutant fly expressing *UAS-GCaMP6s* in PNs. Gray-scale images represent the AL structure highlighting the attractant- (DM1, DM2, DM3, and DM4) and repellent-responsive glomeruli (DL1 and DL5) with colored circles. Calcium responses are shown to stimulation with ethyl acetate ($10^{-3}$), benzaldehyde ($10^{-1}$), and their binary mixture (MIX(−)). Scale bar = 20 μM. **c** Mean PN activity of repellent- and attractant-responsive glomeruli during stimulation with ethyl acetate ($10^{-3}$, bright blue-green), benzaldehyde ($10^{-1}$, red), and MIX(−) (orange) in Or10a$^{-/-}$ mutant flies. Individual flies are given by individual dots and lines; mean is indicated by black thick line ($n = 10$, paired $t$-test). Pairwise comparisons of mixture responses to the response with the strongest single component (i.e. ethyl acetate or benzaldehyde) are shown for each animal. **d** Schematic of the experimental design: Or7a-expressing ORNs (targeting DL5) are not functional in a Or7a$^{-/-}$ mutant background. Color code indicates glomerulus-specific activation by the attractive (bright blue-green) or repellent (red) odor. **e** Representative gray-scale and pseudocolored images of odor-evoked calcium responses in PNs from three focal planes of a Or7a$^{-/-}$ mutant fly expressing *UAS-GCaMP6s* in PNs. Calcium responses are shown to stimulation with ethyl acetate ($10^{-3}$), benzaldehyde ($10^{-1}$), and their binary mixture (MIX(−)). Scale bar = 20 μM. **f** Mean PN activity of repellent- and attractant-responsive glomeruli during stimulation with ethyl acetate ($10^{-3}$, bright blue-green), benzaldehyde ($10^{-1}$, red), and MIX(−) (orange) in Or7a$^{-/-}$ mutant flies ($n = 12$, paired $t$-test). **g** Schematic of the T-maze assay. **h** Box plots showing behavioral preference indices in the T-maze of Or10a$^{-/-}$ mutant (pink), Or7a$^{-/-}$ mutant (purple), and w1118 flies (gray) to the odors benzaldehyde ($10^{-1}$), ethyl acetate ($10^{-2}/10^{-3}$), MIX(+), and MIX(−) against the solvent control (MOL). ($n = 15$–19, one-way ANOVA with posthoc Tukey test, **$p < 0.01$). Filled boxes are significantly different from zero, empty boxes not (Student's $t$-test)

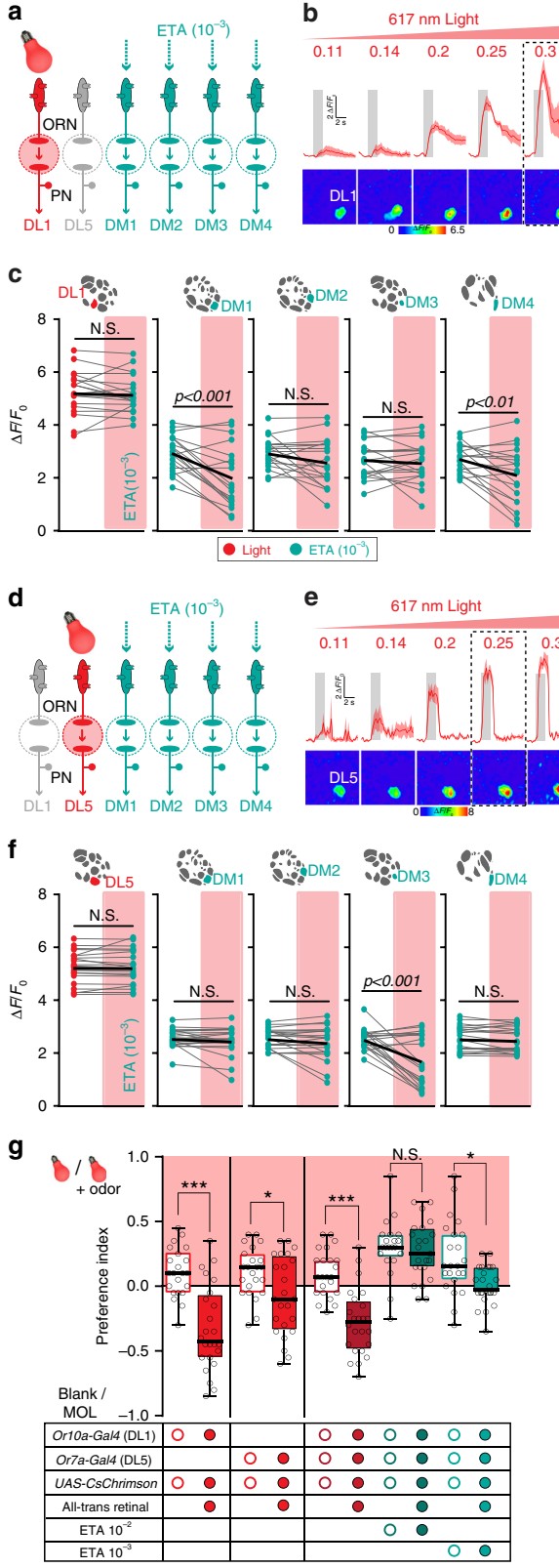

**Fig. 5** Optogenetic activation of repellent-responsive glomeruli unveils glomerular crosstalk. **a** Schematic of the experimental design: artificial activation of CsChrimson by red light in Or10a-expressing ORNs (targeting DL1) during stimulation with ethyl acetate. Color code reflects activation by light (red) or attractive odor (bright blue-green). **b** Calcium signals (time traces and pseudocolored images) of glomerulus DL1 to photoactivation with increasing intensities of 619 nm light for 2 s. Gray boxes indicate light stimulation. Lines represent averaged response, shadows give SEM ($n = 3$). Dashed box marks the light intensity used for further experiments. **c** Mean PN activity of repellent- and attractant-responsive glomeruli during stimulation with either light (red dots), ethyl acetate ($10^{-3}$, bright blue-green dots), or both combined (additional red rectangles) in flies expressing CsChrimson in ORNs of DL1 and GCaMP3 in PNs ($n = 19$, paired $t$-test). **d** Schematic of the experimental design: artificial activation of CsChrimson by red light in Or7a-expressing ORNs (targeting DL5) during stimulation with ethyl acetate. **e** Same experiment as in **b** but for CsChrimson expression in ORNs of DL5 ($n = 4$). **f** Same experiment as in **c** but for CsChrimson expression in ORNs of DL5 ($n = 21$, paired $t$-test). **g** Box plots showing preference indices in the T-maze assay of flies with artificial activation of ORNs expressing Or10a (DL1), Or7a (DL5) or both via CsChrimson by red light alone (bulb) or red light combined with an odor (bulb + odor) against the dark arm of the T-maze without (Blank) or with solvent (MOL) ($n = 22-24$, one-way ANOVA with posthoc Sidak's multiple comparisons test, $*p < 0.05$, $***p < 0.001$). Treatment and genotypes are indicated by the table below. In our assay, control flies (no all-trans retinal) showed reproducible slight attraction to light. Based on this, we used the control flies as the comparison point for calculating optogenetically driven avoidance

---

**Mixture inhibition is mediated by GABA.** We next turned our attention to the neuronal mechanism. Most of the odor-induced inhibitions in the *Drosophila* AL are mediated by the inhibitory neurotransmitters GABA, which binds to $GABA_A$ and $GABA_B$ receptors[12,27,28], and glutamate, which opens glutamate-gated chloride channels (GluClα)[39].

In order to block GABAergic and/or glutamatergic receptors in the AL we applied the antagonist CGP54626 (50 μM) to silence $GABA_B$ receptors and picrotoxin (100 μM) to block the Rdl subunit of the $GABA_A$ receptor and the GluClα. We simultaneously monitored the odor-induced calcium signals in PNs. By blocking $GABA_B$ receptors, we noticed a reduction in the MIX(−)-induced inhibition in the four attractant-responsive glomeruli compared to the saline or wash-out situation (Fig. 6a, b). To quantify this reduction, we calculated the differences between the normalized peak responses upon stimulation with MIX(−) and the attractant alone (Fig. 6c). As expected, the peak response differences in the four attractant-responsive glomeruli were significantly reduced after CGP54626 treatment compared to the controls. Interestingly, after blocking $GABA_A$ and GluClα receptors, only glomeruli DM1 and DM4 showed a significant reduction in their inhibition to MIX(−) (Fig. 6d–f). Picrotoxin could not be washed out, as shown previously[15]. When we applied both antagonists simultaneously, the MIX(−)-induced inhibition was totally abolished in all four attractant-responsive glomeruli (Fig. 6g–i), while we did not observe any obvious effects on the repellent-responsive glomeruli (Supplementary Fig. 6).

Our pharmacological approach has two weak points: first, picrotoxin at the used concentration blocks both, the $GABA_A$ and GluClα receptors. Second, the antagonists act on the pre- and postsynaptic sites and do not allow pinpointing where the inhibition takes place. To overcome these issues we used RNA interference to target either GABAergic or glutamatergic receptors selectively at the pre- and postsynaptic sites of AL input and output neurons. We employed *UAS-Rdl RNAi* against

---

compared to the control flies (Fig. 5g). This supports the notion that these repellent-responsive glomeruli function as aversive input channels and belong to a special subset of valence-specific glomeruli. This assumption is further substantiated by the fact that the PN axons of DL1 and DL5 in the lateral horn reveal very similar and overlapping axonal arborizations as other aversive-specific PNs[38].

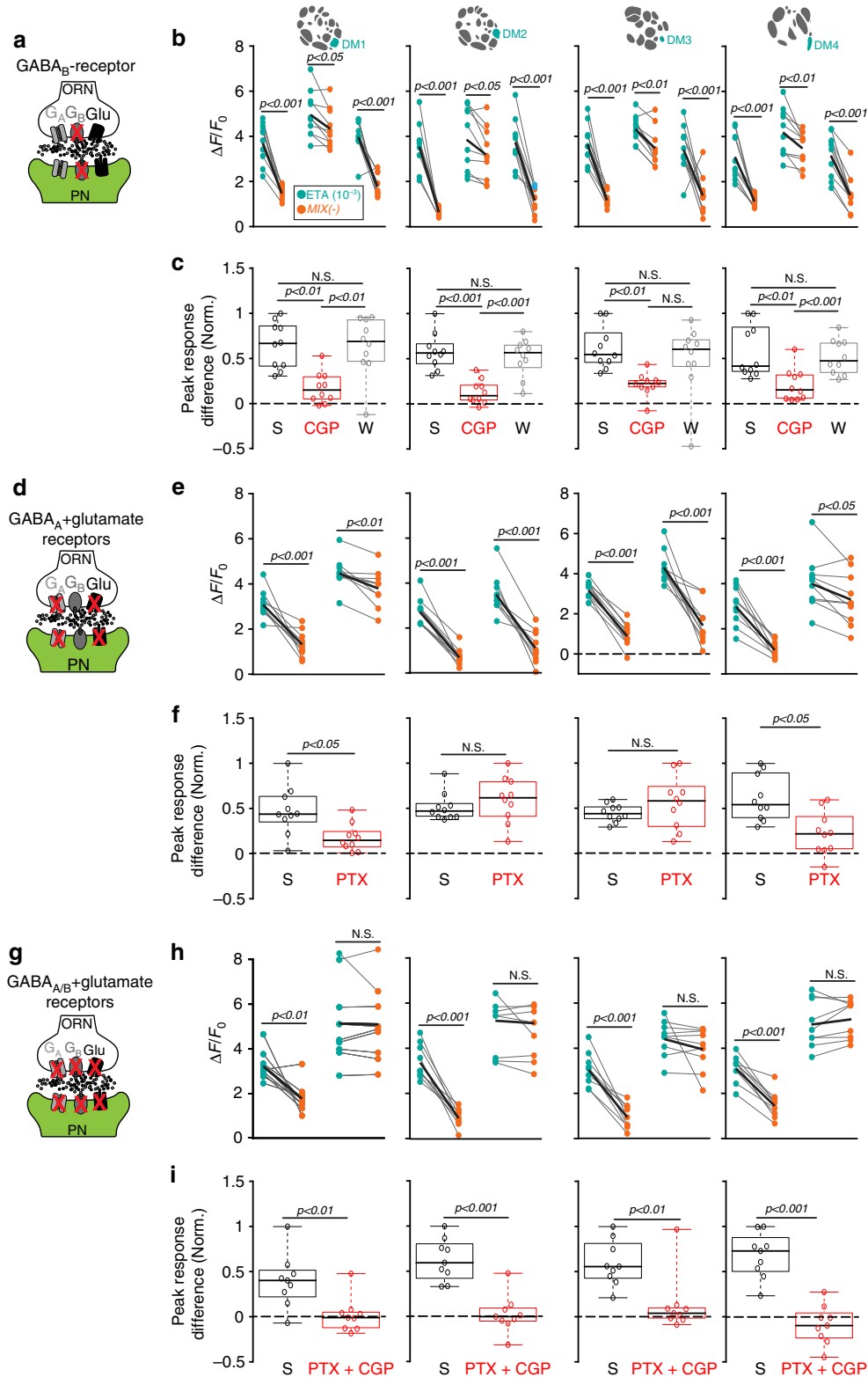

the Rdl subunit of GABA$_A$[40], *UAS-GBi* against the GABA$_B$R2 subunit[28], *UAS-gluclα* RNAi against the GluClα[39] and *UAS-empty-RNAi* as a control. We confirmed the efficiency of the RNAi lines by RT-PCR (Supplementary Fig. 7a). First, we expressed these RNAi lines separately at the postsynaptic sites (i.e. in PNs), while visualizing odor-evoked calcium signals in PNs (Fig. 7a). Interestingly, blocking GABA$_A$ receptors significantly reduced the inhibition to MIX(−) in two out of four

attractant-responsive glomeruli (DM1 and DM4) (Fig. 7b, c). Neither silencing GABA$_B$-receptors nor GluClα did affect the inhibition induced by MIX(−) in any of the attractant-responsive glomeruli. The repellent-responsive glomeruli revealed a linear mixture response independent of the RNAi line expressed (Supplementary Fig. 7b). As expected, the representation of MIX(+) was also not influenced by any RNAi expression (Supplementary Fig. 7c). These findings indicate that GABA$_A$

**Fig. 6** MIX(−)-induced inhibition can be blocked by GABA- and glutamate antagonists. **a** Schematic illustrating the experimental design: GABA$_B$ antagonist CGP54626 (50 μM) is applied while calcium responses of PNs are monitored (green). **b** Mean PN activity of the four attractant-responsive glomeruli showing the effect of CGP54626 (CGP) compared to saline (S) and wash-out (W) during stimulation with ethyl acetate ($10^{-3}$, ETA, bright blue-green) and MIX(−) (orange). Individual flies are given by single dots and lines; mean is indicated by black thick line (n = 10, paired $t$-test). **c** Box plots represent normalized peak response differences of odor responses of the glomeruli shown in **b**. Differences were calculated by subtracting calcium signals to MIX(−) from those to ethyl acetate during different treatments (i.e. 1 represents strongest mixture inhibition, while 0 means no inhibition). Circles show individual animals (n = 10, one-way ANOVA with Bonferroni's multiple comparisons test). **d** Schematic illustrating the experimental design: GABA$_A$ and glutamate antagonist picrotoxin (100 μM) is applied while calcium responses of PNs are monitored (green). **e, f** Same representations as in **b, c** for picrotoxin (PTX) compared to saline (S) (n = 10, Student's $t$-test). **g** Schematic illustrating the experimental design: mixture of CGP54626 (50 μM) and picrotoxin (100 μM) is applied to block GABA$_A$, GABA$_B$, and glutamate receptors while calcium responses of PNs are monitored (green). **h, i** Same representations as in **b, c** for the combined application of picrotoxin and CGP54626 (PTX + CGP) compared to saline (S) (n = 9, Student's $t$-test)

receptors mediate the MIX(−)-specific inhibition at the post-synaptic site in two out of four attractant-responsive glomeruli, which is well in line with our results deriving from the pharmacological treatment with picrotoxin (Fig. 6d–f).

Although our results show that pharmacological blocking of GABA$_B$ receptors significantly reduced the MIX(−)-induced inhibition in all four attractant-responsive glomeruli (Fig. 6a–c), genetic silencing of GABA$_B$ via RNAi selectively in PNs did not affect the mixture inhibition (Fig. 7a–c). We therefore wondered whether part of the observed GABA$_B$-dependent inhibition derives from inhibition in ORNs. Indeed, GABA$_B$-mediated inhibition at the presynaptic sites has already been well characterized[27,28]. To selectively block GABA$_B$ receptors at the presynaptic site, we expressed the same RNAi lines in ORNs while we monitored the calcium responses in PNs (Fig. 7d). Notably, the MIX(−)-induced inhibition was abolished in DM2 and DM3 and strongly suppressed in DM1 and DM4 when GABA$_B$ receptors in ORNs were silenced (Fig. 7e, f). We did not observe any effects on the odor-evoked responses in the repellent-responsive glomeruli by the RNAi lines, nor did we see any changes in the representation of MIX(+) (Supplementary Fig. 7d, e). In sum, these data show that DM1 and DM4 are inhibited by MIX(−) via GABA$_B$ and GABA$_A$ receptors on both, the pre- and post-synaptic loci, while MIX(−) inhibits the other two attractant-responsive glomeruli (DM2 and DM3) predominantly on the pre-synaptic site via GABA$_B$ receptors.

As part of the inhibition of the attractant-responsive glomeruli is mediated by GABA$_B$ receptors in ORNs, we next verified that the MIX(−)-induced inhibition occurs at the sensory level and performed calcium imaging in ORNs. As expected, stimulation with MIX(−) induced a strong and significant inhibition in the attractant-responsive glomeruli, while MIX(+) was linearly represented (Supplementary Fig. 8a, b).

Coincidentally, the two different ORN types that detect ethyl acetate (i.e Or42b/DM1) and benzaldehyde (i.e. Or10a/DL1) are housed in the same basiconic sensillum ab1[8,24]. As non-synaptic inhibition between different ORN types located in the same sensillum has been shown in the *Drosophila* antennae[4], we performed single-sensillum recordings from the ab1 sensillum to exclude that part of our observed mixture inhibition derives from peripheral interactions. However, we did not observe any mixture inhibition at the sensillum level (Supplementary Fig. 8c, d), implying that the MIX(−)-induced inhibition derives from inhibitory interactions within the AL network.

In conclusion, our results demonstrate that GABA mediates the observed mixture inhibition in the attractant-responsive glomeruli. Glomeruli DM2 and DM3 are inhibited at the presynaptic locus through the GABA$_B$ receptor, while DM1 and DM4 are inhibited at their pre- and postsynaptic terminals via GABA$_B$ and GABA$_A$ receptors, respectively. To the best of our knowledge, our results are novel in revealing that specific glomeruli are inhibited at two different synaptic

sites, while other glomeruli are inhibited solely at the presynaptic locus.

**Defined subsets of GABAergic LNs mediate mixture inhibition.** Finally, we aimed at identifying the LN population underlying the mixture-induced GABAergic inhibition. We selected four different enhancer trap lines that label various types of GABAergic LNs ranging from pan-glomerular, continuous, and regional to patchy LN populations[13]. To selectively silence LN-mediated inhibition we expressed an RNAi construct against glutamic acid decarboxylase (Gad) in conjunction with *UAS-Dicer2* to knock-down GABA synthesis[41] in each of the different LN subsets, while we monitored calcium responses to MIX(−) and the individual odors in PNs. We confirmed the reduction in GABA production via immunostaining (Fig. 8a–d). Interestingly, silencing GABA release in two out of the four LN lines, that label many patchy LNs[13], significantly reduced the mixture inhibition in the attractant-responsive glomeruli. The mixture inhibition in DM3 was reduced by silencing GABAergic LNs using *NP3056-Gal4* (Fig. 8b and Supplementary Fig. 9a), while knocking-down GABA release in *HB4-93-Gal4* abolished the mixture inhibition in DM1 and DM4 (Fig. 8d and Supplementary Fig. 9a). In contrast, GABAergic LNs labeled by the mostly pan-glomerular LN lines *GH298-Gal4* and *H24-Gal4*[13] seem not to contribute to the inhibition of the attractant-responsive glomeruli (Fig. 8a, c). Moreover, we observed that the response to ethyl acetate was increased in flies where the GABA release was silenced in *GH298-Gal4* and *NP3056-Gal4* (Supplementary Fig. 9a), which might be due to the absence of gain control. We attempted to monitor the behavioral consequences regarding mixture processing of flies with impaired lateral inhibition. However, such broad manipulations led to unexpected behavioral responses to single odors which prevented us from employing these RNAi lines for further behavioral experiments.

In sum, our results demonstrate that defined LN types are involved in a glomerulus-specific lateral inhibition induced by odor mixtures. Furthermore, our data suggest that *HB4-93-Gal4* should comprise LNs innervating glomerulus DL1 that target glomeruli DM1/DM4, while *NP3056-Gal4* should include LNs that innervate DL5 and inhibit DM3. To confirm that the aforementioned glomeruli are connected by patchy LNs from the corresponding lines, we performed neural tracing by expressing photoactivatable GFP (PA-GFP)[42]. We illuminated PA-GFP in single somata to selectively label individual LNs, reconstructed and annotated their innervation in our glomeruli of interest (Fig. 8e, f). To investigate whether the pre- and postsynaptic densities of those LNs vary between the attractant- and repellent-responsive glomeruli, we expressed synaptotagmin-hemagglutinin (Syt-HA) as a presynaptic marker and dHomer fused with GCaMP3 as a postsynaptic marker[43] selectively in

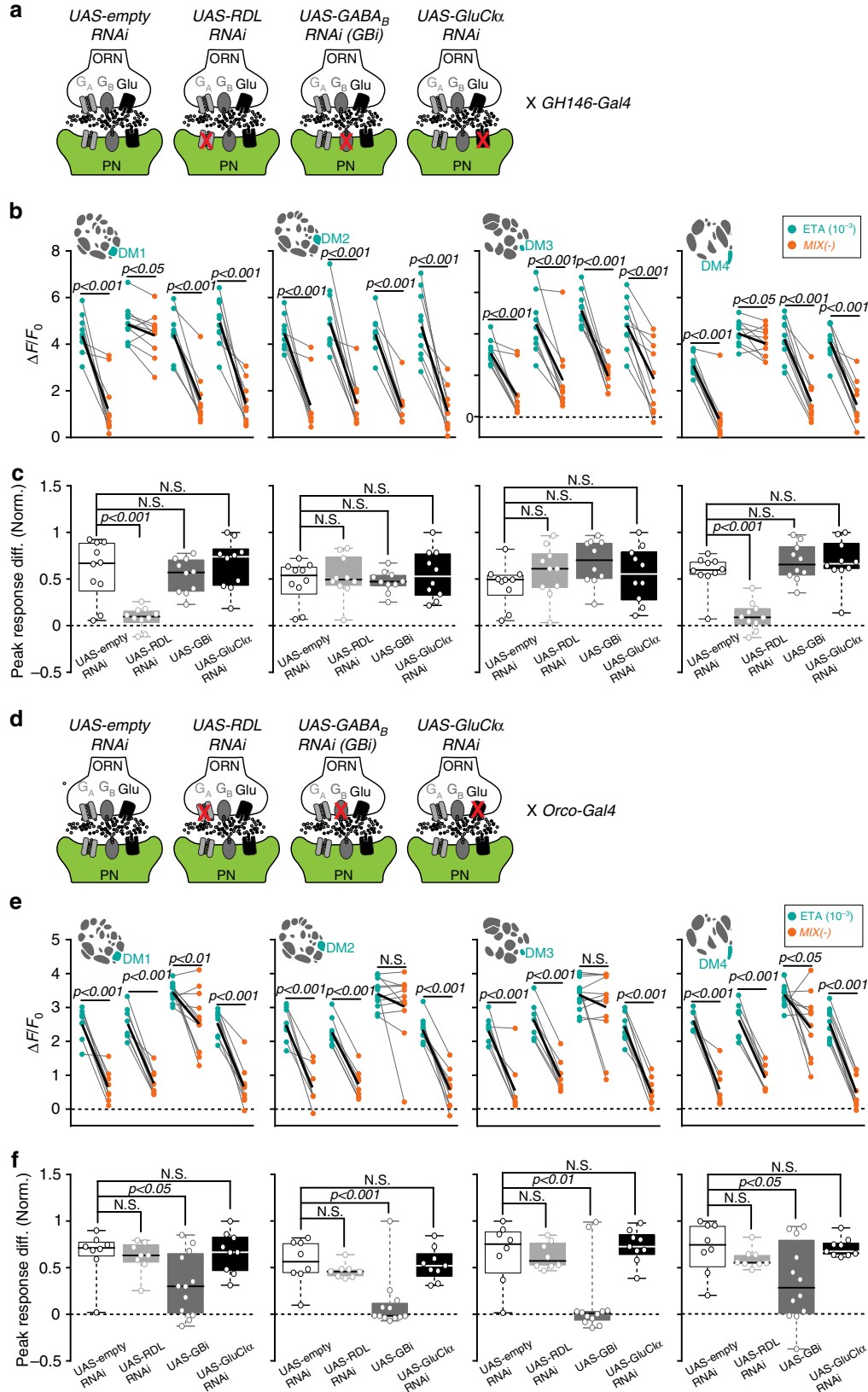

the two LN populations (Supplementary Fig. 9b). Indeed, quantification of the fluorescent signal of both markers reveals that LNs labeled by *NP3056-Gal4* and *HB4-93-Gal4* possess a significantly higher density of postsynapses in the repellent-responsive glomeruli (DL5 and DL1), while presynapses are stronger pronounced in the attractant-responsive glomeruli (DM3 and DM1/DM4; Fig. 8g, h). This data suggests that pre- and postsynapses of inhibitory LNs are not uniformly distributed among different glomeruli which might cause the observed heterogeneity of the lateral inhibition.

**Fig. 7** Mixture inhibition takes place at pre- and postsynaptic sites. **a** Schematic illustrating the effect of different RNAi-lines used to block GABA$_A$, GABA$_B$, or glutamate receptors selectively in PNs, while odor responses in PNs are monitored with 2-photon imaging (green). *UAS-empty-RNAi* serves as the control. **b** Mean PN activity of the four attractant-responsive glomeruli during stimulation with ethyl acetate ($10^{-3}$, bright blue-green) and MIX($-$) (orange) in flies expressing the indicated RNAi lines (**c**) in PNs. Individual flies are given by single dots and lines; mean is indicated by black thick line ($n = 10$, paired *t*-test). **c** Box plots represent normalized peak response differences of odor responses of the glomeruli shown in **b**. Circles show individual animals ($n = 10$, one way ANOVA with Bonferroni's multiple comparisons test). **d** Schematic illustrating the effect of different RNAi-lines used to block GABA$_A$, GABA$_B$, or glutamate receptors selectively in ORNs, while odor responses in PNs are monitored with 2-photon imaging (green). *UAS-empty-RNAi* serves as the control. **e** Mean PN activity of the four attractant-responsive glomeruli during stimulation with ethyl acetate ($10^{-3}$, bright blue-green) and MIX($-$) (orange) in flies expressing the indicated RNAi lines (**f**) in ORNs. Individual flies are given by individual dots and lines; mean is indicated by black thick line ($n = 8$–12, paired *t*-test). **f** Box plots represent normalized peak response differences of odor responses of the glomeruli shown in **e**. Circles show individual animals ($n = 8$–12, one way ANOVA with Bonferroni's multiple comparisons test)

## Discussion

In this study, we analyzed the integration of binary odor mixtures of opposing hedonic valences and demonstrate how glomerular-specific inhibition and crosstalk results in an appropriate behavioral output. We show that glomeruli that strongly respond to the attractive odor are inhibited by the repellent odor in the mixture, which is mediated by defined subsets of GABAergic LNs (Fig. 9). Heterogeneity in responses to mixtures has been shown in previous studies where excitation of some glomeruli by one of the mixture components can inhibit the glomeruli activated by the other component[3]. Similar to invertebrates, evidence for non-linearity of mixture interactions has been reported in individual mitral/tufted cells (PNs analogs) in the olfactory bulb of vertebrates[1,44]. As an alternative scenario it is also conceivable that instead of inhibiting the attractant-coding pathway to shift the behavior towards aversion, the response of the repellent-responsive glomeruli could be boosted via lateral excitation[45]. Lateral excitation has been described to drive synergistic interaction between the binary mixture of cis-vaccenyl acetate and vinegar[26]. Although odors representing sex and food are mutually reinforcing[26], a binary mixture of odors with opposing valences means a conflicting input. We therefore postulate that, in contrast to reinforcing input, conflicting sensory input is processed via lateral inhibition in the fly AL. An assumption that would be intriguing to be tested in the future.

We did not observe any inhibition of the attractant-responsive glomeruli when we stimulated with MIX($+$). This lack of inhibition is probably due to the strong ORN input leading to high presynaptic firing rates in the attractant-responsive glomeruli. Consequently, lateral inhibition deriving from the aversive circuit has only a low impact and does not decrease the excitation of the attractant-responsive glomeruli[5].

Obviously not all glomeruli that are activated by an attractive odor are inhibited by a repellent in a mixture and might not contribute to the attractiveness of an odor. This observation makes sense in the light of accumulating evidence suggesting that the innate behavioral output is correlated either to the summed weights of specific activated glomeruli[22,34] or to the activity of single processing channels[33,36,46,47]. The latter argument is supported by the finding that only very few, special glomeruli seem to be valence-specific and induce clear attraction or aversion behavior upon artificial activation.

It is important to mention that our subset of repellent-responsive glomeruli does also respond to non-aversive and even partly attractive odors, such as E2-hexenal and ethyl benzoate[24]. However, an attractive odorant may indeed activate some aversive input channels beside their main activation of the attractive circuitry (or the other way around). What actually matters is the behavioral output that is consequently elicited when a specific glomerulus becomes activated. For example, ORNs that respond to $CO_2$ are also activated by ethyl benzoate and E2-hexenal[48]. However, the $CO_2$ circuit has been clearly demonstrated to

mediate behavioral aversion[47,49]. Following this argument, artificial activation of glomeruli DL1 and/or DL5 leads to aversive behavior, while silencing DM1 and DM4 abolished attraction to the attractant. These experiments provide evidence that activation of the repellent- and attractant-responsive glomeruli causes a valence-specific behavior, and can therefore be defined as attractive or aversive input channels, respectively.

Interestingly, we observed one exception in our data set: although the repellent odor geosmin reduced the attraction to balsamic vinegar in the mixture, we did not observe any mixture inhibition. The detection of geosmin is one of the rare cases, where an odor is detected by only one receptor type and consequently activates only one glomerulus. Similar specialized pathways have been described for the detection of sex pheromones and $CO_2$[47,50]. Glomeruli processing these ecologically labeled lines differ from broadly tuned glomeruli with regard to their neuronal composition[51]. Hence, it is conceivable that the narrowly tuned geosmin-responsive glomerulus does not exhibit strong interglomerular interactions and has therefore a different impact on the attractant-responsive glomeruli. Mixture interactions between geosmin and attractive odors might be implemented in higher processing centers which contain circuit elements mediating interactions between odors[52].

Lateral inhibition, which is believed to enhance contrast and to facilitate discrimination of similar stimuli, is an important motif throughout the nervous system[53–55]. In mice, dense center-surround inhibition refines mitral cell representation of a glomerular map[56], while other evidence showed that lateral inhibition can be rather selective and biased between different mitral cells[57]. In accordance with the olfactory bulb, the AL exhibits broad, selective or even both forms of lateral inhibition[3,5,15,58], whereby certain glomeruli can show different sensitivities towards an inhibitory input[14]. Lateral inhibition in the *Drosophila* AL is largely mediated through GABA[12,58]. Most of the GABAergic inhibition in the *Drosophila* AL has been shown to take place predominantly on the presynaptic site mediated through GABA$_A$ and GABA$_B$ receptors[27,28]. In addition, PNs also receive GABAergic inhibition via GABA$_A$ and/or GABA$_B$ receptors from LNs[12]. Notably, we found that two out of four attractant-responsive glomeruli are inhibited on the pre- and postsynaptic levels (via GABA$_B$- or GABA$_A$-receptors), while the other two glomeruli are inhibited only presynaptically through GABA$_B$-type receptors. Previous results have so far shown that GABA$_A$-type receptors contribute weakly to lateral inhibition and shape the early phase of odor responses[12]. However, our data demonstrate that GABA$_A$-type receptors largely mediate mixture-induced inhibition during the full period of the odor presentation which is reminiscent to tonic inhibition in the mammalian system[59].

We show that mixture-induced lateral inhibition of the attractant-responsive glomeruli was abolished when we silenced GABA synthesis in mostly patchy LNs. Hence our data suggest, in consistency with previous studies, that LNs with more selective

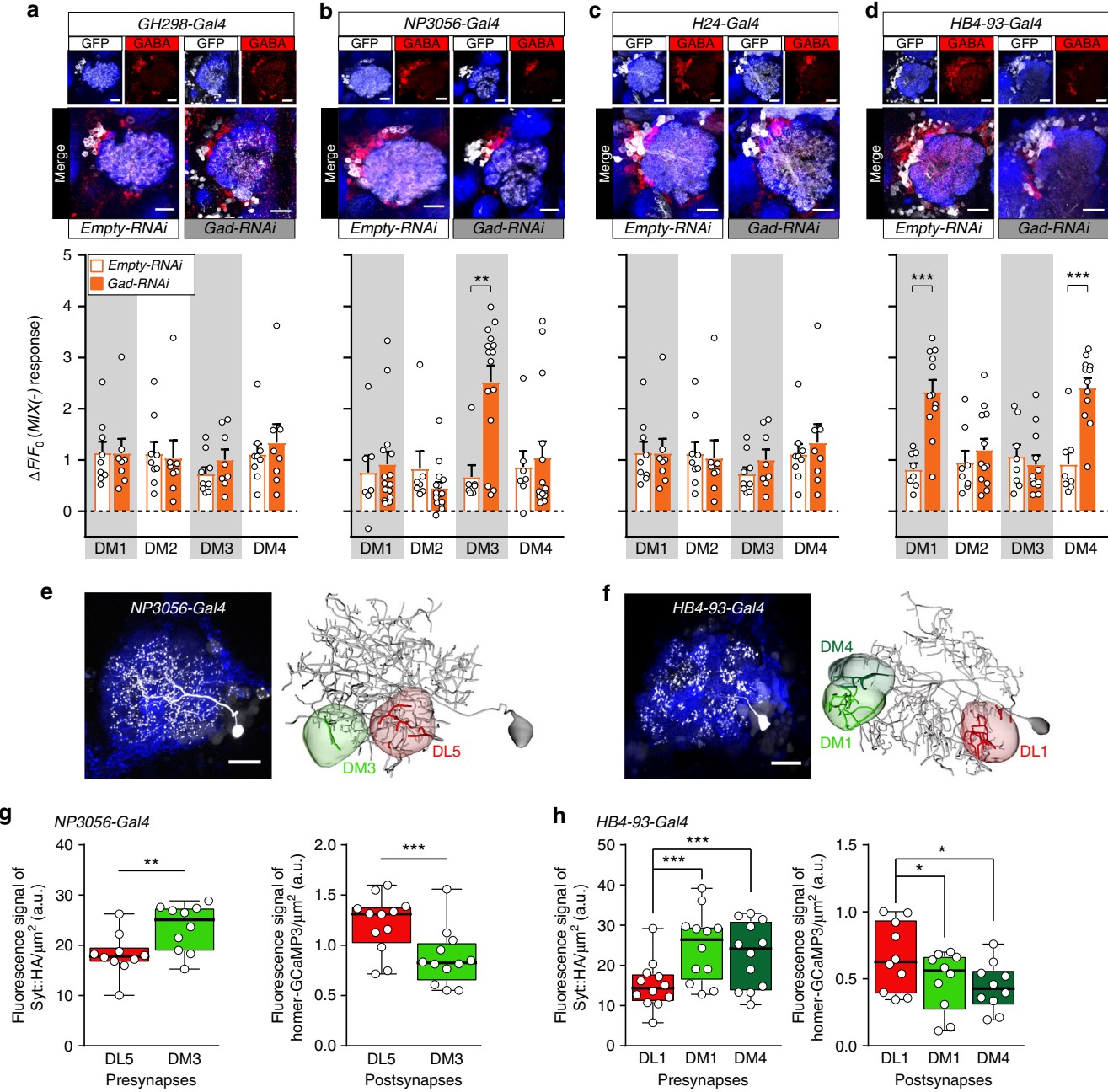

**Fig. 8** Defined subsets of GABAergic LNs mediate mixture inhibition. **a–d** Upper panel, immunostaining against GFP (white) and GABA (red) within the AL of four different Gal4 lines that label LN subpopulations with intact (empty-RNAi) or silenced (Gad-RNAi) GABA production. Scale bar = 20 μm. Lower panel, barplots represent mean PN activity of the four attractant-responsive glomeruli during stimulation with MIX(−) in flies with intact or silenced GABA production in four different LN subpopulations (shown in **a–d**). Individual flies are given by individual dots. Data are represented by mean+SEM ($n = 7$-15, Student's $t$-test, ** $p < 0.01$, *** $p < 0.001$). Silencing GABA release of LNs labeled by *NP3056-Gal4* results in a strong MIX(−) response in PNs of glomerulus DM3, while glomeruli DM1 and DM4 show strong mixture responses when *HB4-93-Gal4* LNs were silenced. This response increase to MIX(−) indicates a relief of the mixture-induced inhibition. **e, f** Left, representative individual patchy LNs, labeled by photoactivating PA-GFP in single somata of *NP3056-Gal4* LNs (**e**), and *HB4-93-Gal4* LNs (**f**). To facilitate glomerular identification, *GH146-QF, QUAS-mtdtomato* (blue) was expressed. Scale bar 20 μm. Right, neuronal reconstructions of two exemplary single LNs. Glomeruli that are supposed to be connected by these LNs are highlighted. **g** Boxplots representing the quantification of the florescence signal of the presynaptic marker *UAS-Syt::HA* (left panel) and the postsynaptic marker *UAS-homer-GCaMP3* (right panel) expressed under control of *NP3056-Gal4* ($n = 10$ for pre-, $n = 12$ for postsynapses, two-sample $t$-test). **h** Same as in **g** for *HB4-93-Gal4* ($n = 12$ for pre-, $n = 10$ for postsynapses, one-way ANOVA with posthoc Dunnett's multiple comparisons test). * $p < 0.05$, ** $p < 0.01$, *** $p < 0.001$. See Supplementary Fig. 9b, c for immunostaining of these markers

innervations mediate glomerulus-specific interactions and rather contribute to mixture processing, while pan-glomerular LNs (e.g. *GH298-Gal4* and *H24-Gal4*), that globally release GABA, might be involved in gain control[10,12,13,16].

Interestingly, the repellent-responsive glomeruli DL1 and DL5 did not show any mixture interaction, but mediate the lateral inhibition of the attractant-responsive glomeruli. We can think of two possible scenarios that would provide the neuronal

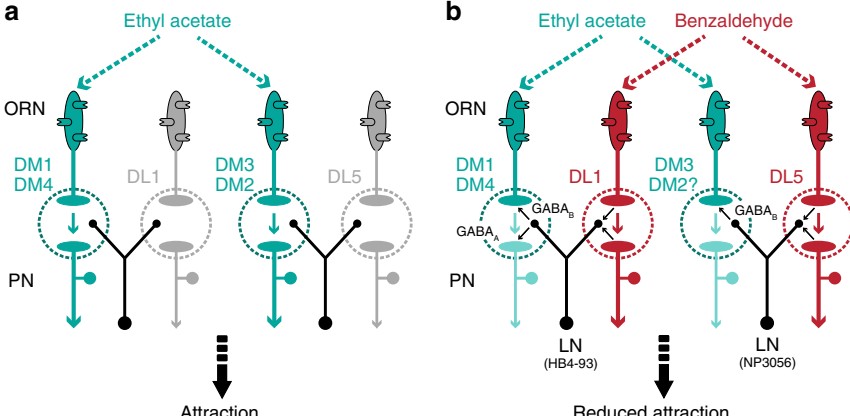

**Fig. 9** Circuit model for glomerulus-specific crosstalk in the fly AL. **a** Stimulation with the attractive odor ethyl acetate activates the attractant-responsive glomeruli DM1, DM2, DM3, and DM4, which results in behavioral attraction. **b** The repellent odor benzaldehyde is blended with the attractive odor ethyl acetate. Benzaldehyde activates the repellent-responsive glomeruli DL1 and DL5 which induce an inhibition of the attractant-responsive glomeruli via two different subsets of GABAergic LNs: DL1 is mediating the inhibition of glomeruli DM1 and DM4 via *HB4-93*-type LNs, while glomeruli DM3 and, to some extent, DM2 are inhibited by DL5 via *NP3056*-type LNs. This inhibitory crosstalk shifts the mixture representation towards the repellent odor and consequently leads to a reduced behavioral attraction

substrate for this mechanism dependent on either the donor (i.e. LNs) or the receiver (i.e. glomerulus) side. First, since glomeruli vary dramatically in their GABA sensitivity and consequently their sensitivity to LN activation[14], lateral inhibition is heterogeneous across different glomeruli. Second, lateral inhibition is biased among different glomeruli due to a glomerulus-specific synaptic distribution of pre- and postsynapses of GABAergic LNs, i.e. the GABA release is not uniform. This assumption is supported by our data revealing that GABAergic LNs possess a higher density of postsynapses in DL1 and DL5 than in the attractant-responsive glomeruli. In line with our findings, EM based data from the larvae AL describe GABAergic, oligoglomerular 'choosy' LNs with a clear polarity contributing to postsynaptic inhibition for most glomeruli, while they receive inputs from only a small glomerular subset[60]. Hence, there is strong evidence that some glomeruli can drive lateral inhibition in other glomeruli. Both scenarios could either occur separately or reinforce each other. Moreover, it might be ecological relevant not to inhibit the input of the aversive pathways since these are associated with life-threatening situations that should be coded reliably and rather override an attractive input[33,60].

In contrast to our expectation, sole photoactivation of DL1 or DL5 or stimulation with the repellent alone did not induce inhibition in the attractant-responsive glomeruli. This might be due to the low spontaneous activity of ORNs innervating the attractant-responsive glomeruli[24], which correlates with spontaneous fluctuations in the membrane potential of the postsynaptic PNs[61]. Consequently, inhibitory responses (i.e. hyperpolarizations) are difficult to capture with calcium imaging.

In other sensory systems, lateral inhibitory connections of neuronal subsets involved in sensory processing have been elucidated in great detail, such as in the retina of mice[62] or the rat visual cortex[63]. Also for the *Drosophila* AL, previous studies suggested that glomerular subgroups are connected via inhibitory LNs[3,15,58]. However, these studies could neither pinpoint the precise connections nor their significance for behavioral perception. Our data provide evidence for a specific inhibitory crosstalk between identified glomeruli and substantiate the existence of selective lateral inhibition in the fly AL. Our postulated network circuits offer insights into the principle of sensory integration. It will be intriguing to see whether neuron-specific crosstalk

represents a general phenomenon to integrate multiple and rather conflicting input channels in other sensory modalities.

## Methods

**Fly stocks.** Flies were reared on conventional cornmeal agar medium under 12 h/12 h light/dark cycle at 25 °C (except for the experiment involving PCR, which were reared at 18 °C). All experiments were performed on adult females. Genotypes used in each figure are listed in Supplementary Table 1. The following stocks were used: Canton-S wildtype flies, *GH146-Gal4*[31] (from Leslie Vosshall's lab.), *20XUAS-IVS-GCaMP6s* (attP40)[29] (Ilona Kadow's lab.), *20XUAS-IVS-GCaMP6f* (VK00005) (Bloomington Drosophila Stock Center (BDSC) 52869), w1118 (BDSC 3605), *GH298-Gal4*[31] (Reini Stocker's lab.), *NP3056-Gal4* (Kyoto DGGR 113080), *HB4-93-Gal4* (liqun luo's lab), *H24-Gal4* (BDSC 51632), Or10a mutant (Or10a[f03694])[64] (BDSC 18684, culled July 2017), Or7a mutant (Or7a−/−)[65] (gift from Christopher Potter), *UAS-Rdli-G* and *UAS-GBi*[28,66] (both are gifts from Mani Ramaswami), *UAS-dicer2* (BDSC 24644), *Or42b-Gal4* (BDSC 9972), *Or59b-Gal4* (BDSC 23898), *UAS-Kir2.1* (P(w[ + mC] = UAS-Hsap/KCNJ2.EGFP)1 BDSC 6596), *UAS-Syt::HA; UAS-mCD8-GFP* (gift from Hiromu Tanimoto), *UAS:homer-GCaMP3.0* (gift from André Fiala), *GH146-QF,QUAS-mtdTomato* (BDSC 30037), *UAS-C3PA* (gift from Sandeep Datta), *nsyb-Gal4* (BDSC 51635), *tubP-GAL80*[ts] (BDSC 7018), *UAS-GluClα RNAi*[39] (BDSC 53356), *Or10a-Gal4* (BDSC 9944), *Or7a-Gal4* (BDSC 23908), *UAS-empty RNAi* (attp40) (BDSC 36304), *20XUAS-CsChrimson-mCherry-trafficked* (VK00005) and *20XUAS-CsChrimson-mCherry-trafficked* (in su(Hw) attP5) (both are gifts from Vivek Jayaraman), *Orco-Gal4* (gift from André Fiala), *Orco-Gal4* on (II) (BDSC 26818), *UAS-GAD RNAi* (BDSC 51794), and *GH146-QF, QUAS-GCaMP3.0*[34] (gift from Hokto Kazama's lab.).

**FlyWalk assay.** FlyWalk experiments were performed adapting previous protocols[18,67]. In brief, we tested 15 starved (24 h) female flies in 15 parallel glass tubes (inner diameter 0.8 cm). The flies were continuously exposed to a humidified airstream with a velocity of 20 cm s−1 (20 °C, 70% relative humidity). All experiments were performed under red light background conditions (λ = 630 nm) generated by a LED cluster. Flies were monitored during the whole experiment using an overhead camera (HD Webcam C615, Logitech, Switzerland). Odor stimulation was done using a multicomponent stimulus device, where flies were repeatedly presented 1 s pulses of single odors or mixtures with an interstimulus interval of 90 s. Despite the well-controlled production of the odor stimulus, this stimulus temporarily broadens while moving through the tubing system of the setup. At the same time the stimulus has been shown to decrease in concentration only by <10% along its travel through the system[21] resulting in a ca. Two seconds stimulus of well-defined concentration arriving at the tested flies. At the same time the flies' XY-position was recorded for each pulse. The stimulus protocol consisted of 2 single odors, their mixture, a negative control (mineral oil, MOL) (Carl Roth) and clean air pulses, which were presented for 50 times each in a pseudo-randomized sequence.

Odors were presented to the mixing chamber made of polyetheretherketone (PEEK) via ball-stop check valves. Basically there are two airflows – continuous and odor. The continuous airflow stands for the "no odor"-condition, where the clean airflow passes through empty and clean odor vials. While presenting an odor, the clean airflow is redirected through the vial containing the odor dilution, picking up

the saturated headspace. Thus, odors are presented to the flies with minimal disturbances in the total airflow.

**FlyWalk data analysis**. Since flies are allowed to move freely in the glass tubes, the individuals may have different meeting times with the same odor pulse, depending on whether they sit more upwind or downwind. We corrected this by calculating the encounter for each single fly for each stimulus based on its position, the delay of the odor traveling through the system and the wind speed within the system using a custom-written script in R (https://www.r-project.org/). A second custom-written script was used to calculate the response of the flies towards an odor. On one hand we calculated the mean movement speed of the flies from 1 s before the odor until 7 s after the odor pulse (Fig. 1b, d). Therefore, we analyzed first the average speed within each fly and in the next step we calculated the mean of all flies from the individual averages. When analyzing the upwind displacement (Fig. 1c, e; the distance the flies walking upwind after the odor pulse) we used the same approach, but only within 4 s after the odor pulse. Analysis scripts are available at https://github.com/michathoma/flywalkr.

**2-photon calcium imaging**. All calcium imaging experiments were performed on starved (24 h) female flies aged 4–6 days post-eclosion unless otherwise mentioned. Flies were briefly cold-anesthetized on ice and fixed with the neck onto a custom-made Plexiglas mounting stage with copper plate (Athene Grids, Plano) and a needle before the head to stabilize the proboscis (see also ref. [68]). The head was glued to the stage using Protemp II (3 M ESPE) and the antennae were pulled forward by a fine metal wire. A small plastic plate with a hole that is covered with polyethylene foil was placed on the fly's head. A small cut in the foil was made to expose the head; the foil was sealed to the cuticle using two-component silicon (World Precision Instruments) to prevent the leaking of the Ringer's solution onto the antennae. We added Ringer's solution (NaCl: 130 mM, KCl: 5 mM, MgCl$_2$: 2 mM, CaCl$_2$: 2 mM, Sucrose: 36 mM, HEPES-NaOH (pH 7.3): 5 mM) to the exposed head, and the head cuticle was removed. Care was taken while removing all fat, trachea, and air sacs to reduce light scattering.

ALs were imaged from the dorsal side using a 2-photon laser scanning microscope (2PCLSM, Zeiss LSM 710 meta NLO) equipped with an infrared Chameleon Ultra™ diode-pumped laser (Coherent, Santa Clara, CA, USA) and a ×40 water immersion objective lens (W Plan-Apochromat 40×/1.0 DIC M27). The microscope and the laser were placed on a smart table UT2 (New Corporation, Irvine, CA, USA). The fluorophore of GCaMP was excited with 925 nm. Fluorescence was collected with an internal GaAsP detector. For each individual measurement, a series of 40 frames acquired at a resolution of 256 × 256 pixels was taken with a frequency of 4 Hz. To cover the whole AL, we imaged from 5–6 imaging planes (depending on the preparation) which cover the dorsal-ventral axis of the AL at ~25–30 μm intervals in Figs 2, 3, and Supplementary Figs. 2, 3. Furthermore, no significant odor-evoked signals were observed in between these imaging planes for any of the odors used. In the remaining experiments, we focused on the three focal plans where our main attractant- and repellent-responsive glomeruli (DL1, DL5, DM1, DM2, DM3, and DM4) were accessible. In some cases where the glomeruli's boundaries were not easily detected, we acquired a high-resolution z-stack (1024 × 1024 pixels) at the end of the experiment.

For pharmacological experiments, antagonists were prepared in concentrated stock solutions in Dimethyl sulfoxide (DMSO, Sigma). Just prior to the experiments, we diluted the stock solution into 500 μl Ringer's saline to obtain the final concentration. Picrotoxin (PTX) (Sigma) was prepared as 4.2 mM stock solution in DMSO, while 3-[[1-(3,4-Dichlorophenyl)ethyl]amino]−2-hydroxypropyl](cyclohexylmethyl) phosphinic acid (CGP54626) (Tocris) was prepared as 5.6 mM stock solution in DMSO.

For optogenetic imaging experiment, flies were reared in dark on a standard cornmeal agar food, genotyped under CO$_2$ anesthesia within one day of eclosion. Five males and five females were housed with food supplemented with all trans-Retinal (1.5 mM, Sigma) for 8–9 days. Flies were starved for 24 h on a wet tissue with 5 ml of water supplemented with all trans-Retinal prior imaging. A 619-nm one high power LED collimated with an optic fiber of diameter (4.5 mm) to activate CsChrimson. The optic fiber was placed in front of the antennae on top of the nozzle for the odor delivery. Light powers were measured below the objective (i.e. at the fly's antennae position) using a power-meter (Coherent). Intensity measures ranged from 0.04 to 0.4 mW/mm$^2$. The imaging protocol was the same as described above except we replaced the odor stimulation with 2 s of continuous red light. During imaging, a bandpass emission filter (BP470-550) was used to prevent the 619 nm LED light from interfering with the GCaMP signal and to protect the GaAsP detector. The light onset and offset were triggered using LabVIEW software (National Instruments) which was connected to the ZEN software (Zeiss) and to an LED controller. For "Odor + Light" conditions, light onset was delayed by 0.25 s from the odor onset to account for the delay in odor delivery.

**Odor delivery system for calcium imaging experiments**. We have developed a computer-controlled odor delivery system that is similar to the one used in the FlyWalk to a great extent. Pure compounds were diluted in mineral oil and in water in case of balsamic vinegar. Two millilitre of the diluted odors were added to glass bottle (50 ml, Duran Group, Mainz, Germany), with two sealed openings

for the in-and-out of the air flow (Fig. 2a). For odor application, we used the LabVIEW software (National Instruments) which was connected to the ZEN software (Zeiss) to trigger both image acquisition as well as odor delivery. A continuous airstream, whose flow of 1 l min$^{-1}$ was monitored by a flowmeter. A peek tube guided the airflow to the fly's antennae. For mixtures, the headspaces of the two odors (0.5 l min$^{-1}$ each) were passing through a mixing peek chamber to mix the two headspaces before they were delivered through a common Teflon tube (1 mm diameter) to the fly's antennae resulting in a velocity that with ~20–30 cm s$^{-1}$ was comparable to the airflow in the FlyWalk. In case of single odor, the airflow was compensated by replacing the other odor with clean air (0.5 l min$^{-1}$). Odors were applied during frames 8–15 (i.e. after 2 s from the start of recording for 2 s). 1.5–2 min of clean air was applied between odors, in order to flush any residues of odors and to let the neurons go back to its resting phase.

The flow of the individual odors and the mixtures were monitored by a photo ionization detector (Aurora Scientific) which was placed at the opening of the odor delivery tube. To check whether there are interactions between the two headspaces of the two odors when they were mixed, we performed SPME GC-MS. The SPME fiber was placed in the nozzle of the odor delivery system. Ten times of 2 s odor stimulation was applied on the SPME fiber with 3 s intervals immediately before injection into the GC-MS.

**Immunostaining and microscopy**. Whole-mount immunofluorescence staining was performed following standard procedures[69]. In short, brains were dissected in phosphate-buffered saline (PBS) (Ca$^{+2}$, Mg$^{+2}$ free in room temperature followed by fixing in 4% paraformaldehyde (PFA) in PBS for 30 min at 25 °C. Afterwards, the brains were washed 3–4 times for 1.5–2 h in total in PBS-T (PBS + 0.3% Triton X-100) and blocked for 1 h in PBS-T + 4% normal goat serum (NGS) at 25 °C before incubation in primary antibody diluted in PBS-T + 4% NGS for 48 h at 4 °C. After the incubation period with primary Ab, brains were washed 3–4 times in PBS-T at 25 °C before incubation in secondary antibody for 24 h at 4 °C. After secondary antibodies, brains were washed for 3–4 times for 1.5–2 h at 25 °C in PBS-T before mounted in VectaShield (Vector Labs) on glass slides with bridging coverslips. Stained brains were acquired with Zeiss LSM 880 with a ×40 water immersion objective lens. The following primary antibodies were applied: chicken anti-GFP (1:500, Life Technologies), mouse anti-HA (1:300, Abcam), rat anti-Cadherin (1:30, Developmental Studies Hybridoma Bank [DSHB]), rabbit anti-GABA (1:500, Sigma), and mouse mAb anti-bruchpilot (nc82, 1:30, DSHB); secondary antibodies are Alexa Fluor 488 goat anti-chicken (1:300, Life Technologies), Alexa Fluor 568 goat anti-rabbit (1:300, Life Technologies), Alexa Fluor 633 goat anti-mouse (1:300, Life Technologies).

**Analysis of imaging data**. Functional imaging data were analyzed using a custom written IDL software (ITT Visual Information Solutions) provided by Mathias Ditzen[68], (code available upon request). Each imaging plane was analyzed separately. All recordings were manually corrected for movement. The raw fluorescence signals were converted to $\Delta F/F_0$, where $F_0$ is the averaged baseline fluorescence values of 2 s before the odor onset (i.e. 0–7 frames). For the average $\Delta F/F_0$, average of frames 11–18 was calculated for each trail and averaged among trails. All images were compared with a published in vivo 3D atlas of the AL[30]. The glomeruli could be reliably identified from the baseline fluorescence of GCaMP6s, GCaMP6f, or GCaMP3.0.

To access the strength of the inhibition, we calculated the "peak responses difference" by subtracting the $\Delta F/F_0$ of the mixture from the single odor, and then data was normalized to the maximum value within a glomerulus.

To analysis the presynaptic and postsynaptic signals, brains were dissected and went through the immunostaining procedures. Glomeruli of interest were labeled with a fixed area of region of interest (ROI) using ImageJ and the florescence of either the presynaptic (synaptotagmin-hemagglutinin (Syt-HA)) or postsynaptic (homer-GCaMP 3.0[43]) marker in this ROI was calculated.

**2-photon photoactivation and 3D neuronal reconstruction**. For in vivo photo-activation of individual LNs, 5-day-old female flies (see Supplementary Table 1) were dissected as in the calcium imaging experiments and scanned with a ZEISS LSM 710 NLO confocal microscope (Carl Zeiss, Jena, Germany) equipped with an infrared Chameleon Ultra diode-pumped laser (Coherent, Santa Clara, USA) using a ×63 water immersion objective (W Plan-Apochromat 63×/1.0 VIS-IR, Carl Zeiss, Jena, Germany). A precise region of interest was placed on a single LN soma of the right AL and continuously illuminated for 25–30 min at a wavelength of 760 nm. Subsequently, flies were kept in a dark humidified chamber for approximately 25 min to allow photoconverted GFP molecules to diffuse within the LN. Flies were then killed by removing the body to eliminate movements in the final scan. A z-stack scan of the whole right antennal lobe was acquired at a laser wavelength of 925 nm at an interval of 0.77 μm and with a pixel resolution of 624 × 624. For 3D reconstructions the acquired scans were processed with AMIRA 5.6 software (Fei Visualization Sciences Group) using the labelfield module and the semi-automated reconstruction module "hxskeletonize"[70] to reconstruct glomeruli and single LNs, respectively.

**T-maze experiments**. T-maze experiments were carried out as shown in Fig. 1f. Flies of different genotypes were starved for 24 h before they were tested separately under identical conditions. Odors were presented in the same concentrations as used in the calcium imaging experiments. Instead of a pulsed stimulus the odors in the T-maze apparatus diffused towards direction to the flies. The preference index was calculates as $(O–C)/T$, where $O$ is the number of flies in the odor arm, $C$ is the number of flies in the control arm (i.e. mineral oil arm), and $T$ is the total number of flies used in each trail (i.e. 20 flies). Each trail lasted for 20 min.

For the optogentic experiments, flies were raised in dark throughout their larval stage to adult. Two days after eclosion, flies were genotyped and transferred onto food supplemented with 1.5 mM all-trans retinal for 7 days. Flies were starved on wet kimwipe supplemented with 1.5 mM all-trans retinal for 24 h prior the experiment day. 619 nm LED with 0.3 mW/mm$^2$ was directed on one arm of the T-maze, while the other side was kept in the dark. The preference index was calculated as mentioned above.

**Single sensillum recordings (SSR)**. Three-days-old female adult flies were immobilized in pipette tips, and the third antennal segment or palps were positioned onto a glass coverslip. The tungsten wire electrode (recording electrode) was inserted into extracellular into the base of a sensillum (using a motorized, piezo-translator-equipped micromanipulator (Märzhauser DC-3K/PM-10; http://www.marzhauser.com/de/)) to measure the extracellular signals originating from the ORNs, while the reference electrode was inserted into the eye. Both electrodes were positioned under a microscope (Olympus BX51W1; http://www.olympus.com). Signals were amplified (Syntech Universal AC/DC Probe; www.syntech.nl), sampled (10,667 samples/s), and filtered (100–3000 Hz with 50/60-Hz suppression) via a USBIDAC connection to a computer (Syn-tech). Action potentials were visualized and analyzed using Syntech Auto Spike 32 software. Each measurement was for 10 s, starting 2 s before a stimulation period of 1 s. Responses from individual neurons were calculated as the increase/decrease in the action potential frequency (spikes/s) relative to the prestimulus frequency.

**Expression of receptors (RT-PCR)**. The pan-neuronal driver, neuronal synaptobrevin (nsyb-Gal4), was crossed with UAS-dicer2 and the corresponding UAS-RNAi. Flies were raised at 18 °C. Two-days-old flies were heat shocked at 30 °C for 3 days to relief Gal80 repression before they were moved to 25 °C prior dissection (Supplementary Fig. 7a). Female heads (50–70) were dissected and total RNA was extracted using Tizol (Sigma). Two microgram from each RNA were used to generate the cDNA. RT-PCR was performed using SuperScript One-Step RT-PCR (invitrogen) according to the manufacturer's instructions. Primers for RT-PCR as follow: Rdl-F: 5′-GCG TAT AGA AAA CGA CCT GGT G-3′; Rdl-R: 5′-GGA CAC GAT GCG GTT ATA GTC A-3′; GABABR2-F: 5′-GTA AAG CTC GCC TTG G GT CA-3′; GABABR2-R: 5′-CTG GCC TTG GCT ATG GGA TC-3′; GluClα-F: 5′-CCT ACC TCG CTT CAC ACT GG-3′; GluClα-R: 5′-CCG GTA CTG CTC CTT GAT CC-3′; Rp49-F: 5′-CCA AGA TCG TGA AGA AGC GC-3′; Rp49-R: 5′-CTT CTT GAA TCC GGT GGG CA-3′.

**Statistics**. Sample size was determined based upon preliminary experiments. Statistics were computed using the statistics toolbox in R for the FlyWalk data (https://www.r-project.org/). The rest of the data was statistically analyzed using GraphPad Prism 7 (https://www.graphpad.com/scientific-software/prism/). Graphs were generated using GraphPad Prism 7, R-studio, Excel, or MetaboAnalyst (https://www.metaboanalyst.ca/MetaboAnalyst/faces/home.xhtml) for the PCA. The statistical tests applied for each data set are specified in the corresponding figure legend.

**Reporting summary**. Further information on experimental design is available in the Nature Research Reporting Summary linked to this article.

## Data and Code availability
The data and codes used in this study are available from the corresponding author upon reasonable request.

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

## Acknowledgements

We would like to thank Silke Trautheim for her excellent support for fly rearing, Sascha Bucks for designing primers for the PCR experiment, Michael Thoma for his help and support with the FlyWalk experiments, and Veit Grabe for sharing the schematic of the antennal lobe. We also acknowledge the Bloomington *Drosophila* Stock Center (NIH P40OD018537), Kyoto DGGR, V. Jayaraman, H. Kazama, L. Vosshall, A. Fiala, M. Ramaswami, C. Potter, L. Luo, R. F. Stocker, H. Tanimoto, S. Datta, and I. Grunwald-Kadow for reagents. We thank members of the Sachse and Hansson labs for discussions and comments on the study. The Max Planck Society, the Marie Curie FP7 Programme through FLiACT (A.A.M.M.) and the Alexander von Humboldt Foundation (S.D.C.) supported this work.

## Author contributions

A.A.M.M. performed all functional imaging recordings, T-maze experiments, immuno-histochemistry, molecular genetics, and optogenetics. T.R. performed FlyWalk and SSR experiments. S.D.C. performed immunohistochemistry. B.F. performed photoactivation experiments and neuronal reconstructions. A.A.M.M., S.D.C. and B.F. were supervised by S.S. and T.R. was supervised by M.K. A.A.M.M., B.S.H., M.K. and S.S. together conceived the project. A.A.M.M., M.K. and S.S. designed research, interpreted the results, and wrote the manuscript with input from all authors.

## Additional information

**Competing interests:** The authors declare no competing interests.

