## [Peer Review File · Nature Communications]

Reviewers' Comments:

Reviewer #1:

Remarks to the Author:

In this manuscript Mohamed et al investigated the role of glomeruli specific lateral inhibition in the mediation of mixture interactions across odor representations, and consequently the competition between odor induced attraction and aversion observed in odor mixtures. Authors first identified the specific set of olfactory glomeruli that is involved in suppressed attraction to ethyl acetate via mixing with aversive odor benzaldehyde (Dm1-2-3-4). Later they showed that these 4 glomeruli that are involved in attraction receives differential inhibition from 2 aversive odor responding glomeruli (DI1-5), namely DI1 activation silencing Dm1-4 and DI5 activation silencing Dm3. Authors also showed that this glomeruli specific inhibition of Dm1 and Dm4 is mediated by GABA-A receptors on PNs. Whereas presynaptic inhibition observed at the level of ORNs is prominent in all 4 glomeruli and hence less specific. Finally, the authors showed that reducing GABA release from 2 separate classes of inhibitory LNs with patchy innervation would differentially ablate lateral inhibition on Dm1-4, and Dm3 PNs respectively.

I congratulate the authors for their exciting and novel discovery for showing (for the first time) the presence and function of glomeruli specific lateral inhibition in mediating the fine balance between odor induced attraction and aversion. This work is of relevance to a broad community of people who are interested in the role of lateral inhibition in not only olfaction, but also in other sensory systems. I find that all my major questions coming up while reading the manuscript were answered by the subsequent experiments presented by the authors. The experiments, the statistics and the interpretation of the results are of high quality and the authors used all the state-of-the-art tools of fly behavior, physiology and genetics in a crafty way. Given all these I strongly recommend the publication of this manuscript, after addressing few minor comments that I explain below.

Minor comments:

- 1) Can the authors please discuss why lateral excitation does not exist in those glomeruli responding the aversive odors ? Is this know from before, if not a strong points could be made to highlight this interesting finding, which can be followed up in a future study
- 2) Similarly, these repellent responsive glomeruli do not seem to receive any lateral inhibition, along with my point 1 it appears that these aversive glomeruli seems to be separate from the rest of the antennal lobe. It would be exciting to discuss the ecological meaning of these observations in the discussion section.
- 3) Supplementary figure 5 seems to be too important to leave for supplement. I recommend it to be a main figure if there is space
- 4) Similarly, the data for the control flies, discussed in line 245 and in line 276, is not shown. Please include this in the supplemental data
- 5) Why do the authors think that GH298 and H24 LNs do not contribute to the inhibition of the attractive-responsive glomeruli ? do these lines NOT innervate these 4 glomeruli ? If this is NOT the case can you please discuss why these LN lines labelling many LNs do not contribute to the studies phenomenon.
- 6) In general, I find that the imaging experiments, which are presented in Figure 7 and Figure 6, are extremely convincing. However, I wonder why the authors did not do any behavioral experiments with these lines (after these perturbations) to show that mixture interactions in odor induced behaviors are altered with these manipulations? It is possible that such broad manipulations of PNs, LNs and ORNs would lead to many different un-expected effects in odor induced behavior, which has nothing to do with the mixture suppression of attraction studies here, therefore I would understand if these results do not show the desired effect. However, I find it important that the authors at least discuss why these experiments were not performed in the discussion.

with regards

Emre Yaksi

Reviewer #2:

Remarks to the Author:

Summary:

In the wild, fruit flies would usually encounter odors as mixtures of multiple odorant chemicals at differing concentrations and respond to them appropriately. In this manuscript, the authors propose a mechanism for the neuronal integration of conflicting odor stimuli in the second order, antennal lobe projection neurons. Here, the authors identified mixtures of repellent and attractant odors, titrated such that flies were repelled by the mixtures. They then showed that antennal lobe glomeruli responding to the attractant were inhibited when presented along with the repellent. Specific repellent responding glomeruli were shown to laterally inhibit specific attractant responsive glomeruli by using RNAi knock down and optogenetic activation of subsets of repellent glomeruli. This lateral inhibition was shown to rely on different sub-networks of inhibitory, lateral interneurons within the antennal lobe circuit.

This segregation of the LN network into different groups allowing different aversive glomeruli to inhibit attractive ones and not the reverse is a significant result. This is a very well-presented manuscript with careful neurophysiological measurements supporting an exciting result. It would benefit greatly from some independent, anatomical verification experiments, which I urge the authors to do.

Major comments:

1. Odor stimulus needs clearer characterization: The main result described here depends on the relative concentrations of the attractive odorant in the mix(+) and mix(-) stimuli. The authors have done a very thorough job of characterizing their odor stimuli by doing both, PID and MS measurements. But the current figure supplement 1a does not capture the relative (un-saturated) PID signal amplitudes for ethyl acetate at the two concentrations it was used. This difference in odor vapor concentrations delivered should be measured and depicted, especially because the authors used oil-dilution rather than air-dilution odor-delivery systems. This data probably already exists and just needs to be added to the plot, so this comment should be easy to address.
2. LN sub-population anatomy not characterized for this result: The central, surprising claim in this study is that different LN sub-networks mediate selective, lateral inhibition between sets of glomeruli. There is some anatomical evidence of this in an EM study in larvae, (Berck, ... Cardona, 2016, eLife eLife 2016;5:e14859) and the authors should definitely discuss their findings in light of this. But more importantly, the authors need to provide an independent anatomical basis to their claims. Is it possible to visualize the anatomy of an LN receiving input from an aversive glomerulus and giving output to a subset of the appetitive ones, consistent with the physiology? One possibility is to double label glomeruli and a subpopulation of LNs (as in figure 7) and show that the only the appropriate aversive glomeruli receive dendrites from the labelled LNs. Also show that only the appropriate appetitive glomeruli receive axons.

Minor comments:

1. The authors identify concentrations of attractive odorants and then reduce these to generate the mix(-) stimuli which cause the switch in behavioral response. Did the authors try rather to increase the concentration of the aversive odor to generate the switching mixture?
2. At some points in the manuscript (eg. Line 72) the authors claim to monitor multiple neuronal processing levels, but this is not the case. The model proposed involves glomerular interactions and

while some manipulations involved other levels of the circuit, the level monitored and relevant to the message is that of the glomeruli. I suggest re-focusing the language.

Reviewer #3:

Remarks to the Author:

An interesting, important, and vexing question is how valences of sensory stimuli are determined and encoded by sensory neurons. In "Odor mixtures of opposing valence unveil inter-glomerular crosstalk in the *Drosophila* antennal lobe," authors Mohamed et al. use behavior assays, calcium imaging, optogenetics and other genetic manipulations to test ideas about how attraction and aversion are mediated by defined neural circuits. The manuscript benefits from clear writing, well-designed figures, and abundant data. Unfortunately, the storyline holding all of this together appears to be fundamentally incorrect.

Main concern

The authors identified one set of antennal lobe glomeruli that can be activated by a chemical (ethyl acetate) that causes behavioral attraction, and another set of glomeruli that can be activated by a different chemical (benzaldehyde) that causes behavioral aversion. The authors then varied the valences of their odor presentation by making mixtures of these chemicals that, depending upon the mixture ratio, would cause either attraction or repulsion. The authors then determined how various glomeruli respond as the mixture ratio was varied. The authors interpret their results to conclude that a specific population of GABAergic local neurons mediates interactions between an attraction channel and an aversion channel within the *Drosophila* brain.

The problem is that there is no evidence that attraction and aversion channels really exist. The authors confirm, as previously known, that at tested concentrations, the aversive odorant, benzaldehyde, most strongly activates glomeruli DL1 and DL5 (lines 122ff). "For simplicity (line 124)", the authors call these glomeruli "repellent-responsive." Another set of glomeruli is called "attractant-responsive." At first this nomenclature seems like a reasonable shorthand as it accurately describes the way these glomeruli respond to the very small set of odors tested here. But, as the manuscript progresses, the authors inappropriately begin to take these terms more seriously, for example by repeatedly referring to "aversive neuronal circuitry" and "attractive neuronal circuitry (i.e. lines 188-190)," and suggesting that specific interneurons modify the aversive and attractive actions of generally opposed circuits that mediate behaviors.

This description is internally consistent given the responses to the two or three odorants tested by the authors, but falls apart when considering published data not presented here: the "aversive glomeruli" DL1 and DL5 actually respond best to extremely attractive food and oviposition odors that the authors did not test (such as ethyl benzoate and hexenal, respectively; Laissue and Vosshall, 2008; Lin et al, 2015; etc). Further, odor valence is well-known to be strongly modulated by a variety of factors not tested here, such as satiety level and prior conditioning. Except for the special case of geosmin, the authors present no evidence that general attraction and aversion circuits exist in this part of the brain. Because of this, the authors need to remove this misleading overarching framework from their manuscript.

Absent this general interest framework, what remains is not especially novel or compelling: it is evidence of a specific GABAergic connection between specific glomeruli. The function of this connectivity is known only for the role it plays when triggered by a specific odor; other untested odors, as noted above, would lead to very different results and a very different functional interpretation. The existence of this type of connectivity has long been known (Wilson et al, 2003, for

example, described important aspects of interglomerular crosstalk in *Drosophila*; many other authors working in *Drosophila* and other insects have since described a number of functions served by this crosstalk to transform olfactory representations). Once appropriately rewritten, the results presented here will interest specialists in *Drosophila* olfaction who would like to know about the specific functional connectivity between the known glomeruli revealed here, but will frustrate and mislead readers looking for the broader results suggested by the title, abstract, and discussion of this work.

Specific concerns

(1) line 20: "attractant-responsive input channels"; line 25: "attractant- and repellent-specific circuits", line 228-9: "a specific ORN population that encodes aversive odors", etc, throughout the manuscript: as outlined above, descriptions like these are false and highly misleading.

(2) lines 105-6: "Ethyl acetate and benzaldehyde evoke activity in, mostly, non-overlapping glomeruli." This is true, but they are clearly not separate channels; both odors evoke activity in multiple glomeruli getting input from receptor neurons expressing Or67a, Or85b, Or98a, Or19a (Hallam and Carlson, fig. 6a), so there is some overlap in evoked activity.

(3) lines 124-5: "For simplicity, we name this subset of glomeruli henceforth attractant-responsive or repellent-responsive glomeruli, respectively." This is misleading for two reasons. First, as noted above, the "repellent-responsive glomeruli" also respond very well to attractive odors not discussed in this manuscript. Second, other glomeruli also respond to the test odors (Sup. Fig. 2), so these subsets are incomplete.

(4) line 129: "we noticed strong inhibition in four out of the 34 glomeruli" This result is impossible to interpret because, as the authors themselves correctly note later in the manuscript: "inhibitory responses are difficult to capture with calcium imaging (line 471-2)." It is impossible to say whether other glomeruli also receive inhibitory input.

(5) line 137: "we next examined whether benzaldehyde always inhibits these four attractant-responsive glomeruli when it reduces behavioral attraction in the mixture." This is an overstatement of the evidence: given such a limited stimulus set, the authors cannot conclude what always happens.

(6) line 154-5: "methyl salicylate... which activates only glomerulus DL1" No, methyl salicylate also activates VC2 (Olsen et al, 2007).

(7) around lines 220: The authors suggest that a decrease in activity in DM1-DM4 results in decreased attraction to attractive odorants, but this was not directly tested. Are OR mutants available in which DM1-DM4 activity (or at least DM1&DM4) is selectively decreased in response to the attractants? To strengthen their argument the authors could test such mutants with the Flywalk or Tmaze assays.

(8) lines 222-3: the authors described the data in figure 4f as "silencing glomerulus DL5 and consequently abolishing the inhibition in glomeruli DM2 and DM3". But a substantial difference between activity evoked by ETA and MIX(-) in DM2 remains – inhibition is reduced but not abolished.

(9) lines 224-5: "this suggests that not all glomeruli are crucial for odor valence coding." This is a vast overstatement. The authors tested only two odors. It is perfectly possible that silencing DM5 may have effects on valence coding for other odors. Their evidence shows these glomeruli are not crucial for valence coding under the extremely limited set of conditions tested here.

(10) lines 312-315 and 442-446: The authors argue that GABA_B receptors act pre-synaptically and

GABA_A receptors act post-synaptically in inhibiting the attractant-responsive glomeruli. However, the RNAi experiment that is shown in fig. 6 and sup. fig. 6 used Gal4 lines that are not specific to the ORNs and PNs of interest, leaving open the possibility that inhibition is mediated by other, unknown neurons. The RNAi experiment should be repeated using cell-specific Gal4 lines (for example, Or7a or Or10 gal4 for ORNs).

Response to reviewer comments

Mohamed et al. (NCOMMS-18-21157-T)

REVIEWER #1

(1) Reviewer #1: *In this manuscript Mohamed et al investigated the role of glomeruli specific lateral inhibition in the mediation of mixture interactions across odor representations, and consequently the competition between odor induced attraction and aversion observed in odor mixtures. Authors first identified the specific set of olfactory glomeruli that is involved in suppressed attraction to ethyl acetate via mixing with aversive odor benzaldehyde (DM1-2-3-4). Later they showed that these 4 glomeruli that are involved in attraction receives differential inhibition from 2 aversive odor responding glomeruli (DI1-5), namely DI1 activation silencing Dm1-4 and DI5 activation silencing Dm3. Authors also showed that this glomeruli specific inhibition of Dm1 and Dm4 is mediated by GABA-A receptors on PNs. Whereas presynaptic inhibition observed at the level of ORNs is prominent in all 4 glomeruli and hence less specific. Finally, the authors showed that reducing GABA release from 2 separate classes of inhibitory LNs with patchy innervation would differentially ablate lateral inhibition on Dm1-4, and Dm3 PNs respectively.*

I congratulate the authors for their exciting and novel discovery for showing (for the first time) the presence and function of glomeruli specific lateral inhibition in mediating the fine balance between odor induced attraction and aversion. This work is of relevance to a broad community of people who are interested in the role of lateral inhibition in not only olfaction, but also in other sensory systems. I find that all my major questions coming up while reading the manuscript were answered by the subsequent experiments presented by the authors. The experiments, the statistics and the interpretation of the results are of high quality and the authors used all the state-of-the-art tools of fly behavior, physiology and genetics in a crafty way. Given all these I strongly recommend the publication of this manuscript, after addressing few minor comments that I explain below.

(1) RESPONSE: Thank you very much for the highly positive feedback! We have thoroughly revised our manuscript according to all reviewers' suggestions and have tried to clarify all points of criticism.

(2) Reviewer #1: *Minor comments: 1) Can the authors please discuss why lateral excitation does not exist in those glomeruli responding the aversive odors ? Is this known from before, if not a strong points could be made to highlight this interesting finding, which can be followed up in a future study.*

(2) RESPONSE: We agree with the reviewer that this is an interesting question and have added this point to our discussion (lines 436-445).

(3) Reviewer #1: *2) Similarly, these repellent responsive glomeruli do not seem to receive any lateral inhibition, along with my point 1 it appears that these aversive glomeruli seems to be separate from the rest of the antennal lobe. It would be exciting to discuss the ecological meaning of these observations in the discussion section.*

(3) RESPONSE: We have added to the discussion two possible scenarios that would provide the neuronal mechanism underlying this observation dependent on either the donor (i.e. LNs) or the receiver (i.e. glomerulus) side. On one hand since glomeruli vary dramatically in their GABA sensitivity and consequently their sensitivity to LN activation (Hong and Wilson, 2015, Neuron), lateral inhibition is heterogeneous across different glomeruli. On the other hand we discuss that the synaptic distribution of pre- and postsynapses of GABAergic LNs can be heterogenic and biased as shown for 'choosy' LNs in the larvae antennal lobe (Berck et al., 2016, eLife). In addition, we have performed additional experiments and present new data that provide evidence that GABAergic LNs of the lines *NP3056-Gal4* and *HB4-93-Gal4* (that were involved in the mixture inhibition) possess a higher density of postsynapses in the *repellent-responsive* glomeruli DL1 and DL5 than in the *attractant-responsive* glomeruli (**new Figure 8g,h**). Furthermore, we briefly speculate about the ecological relevance that the *repellent-responsive* glomeruli are not inhibited (lines 507-525).

(4) Reviewer #1: 3) *Supplementary figure 5 seems to be too important to leave for supplement. I recommend it to be a main figure if there is space.*

(4) RESPONSE: We have moved Suppl. Figure 5 to the main figures (**new Figure 6**) as proposed.

(5) Reviewer #1: 4) *Similarly, the data for the control flies, discussed in line 245 and in line 276, is not shown. Please include this in the supplemental data.*

(5) RESPONSE: We have included the data of the *repellent-responsive* glomeruli for the pharmacological treatment as proposed by the reviewer (**new Suppl. Fig. 5**). The other data (former line 245) were already included without any reference to the figures in the text, which was now added.

(6) Reviewer #1: 5) *Why do the authors think that GH298 and H24 LNs do not contribute to the inhibition of the attractive-responsive glomeruli ? do these lines NOT innervate these 4 glomeruli ? If this is NOT the case can you please discuss why these LN lines labelling many LNs do not contribute to the studies phenomenon.*

(6) RESPONSE: We propose that global innervating LNs, such as *GH298-Gal4* and *H24-Gal4*, which also innervate the four *attractant-responsive* glomeruli, are rather involved in gain control and might not contribute to specific lateral inhibition. We emphasized this point in the discussion section.

(7) Reviewer #1: 6) *In general, I find that the imaging experiments, which are presented in Figure 7 and Figure 6, are extremely convincing. However, I wonder why the authors did not do any behavioral experiments with these lines (after these perturbations) to show that mixture interactions in odor induced behaviors are altered with these manipulations? It is possible that such broad manipulations of PNs, LNs and ORNs would lead to many different un-expected effects in odor induced behavior, which has nothing to do with the mixture suppression of attraction studies here, therefore I would understand if these results do not show the desired effect. However, I find it important that the authors at least discuss why these experiments were not performed in the discussion.*

(7) RESPONSE: We did indeed perform behavioral experiments with flies bearing RNAi lines against GABA_x receptor. However, as the reviewer predicted, such a global manipulation impaired strongly the behavioral response to individual odors compared to the control flies (which are represented in Fig. 1b,d in the manuscript) as shown in the figure below (**Reviewer Fig. 1**). We therefore decided not to use these flies for behavioral assays. We added this issue to the MS as the reviewer suggested (lines 387-392).

Reviewer Fig. 1: Behavioral responses to the attractive odor ethyl acetate (ETA, blue-green), the aversive odor benzaldehyde (BEA, red) and their binary mixture (*MIX(+)*, yellow) of flies with silenced GABA_A receptors in the PNs (*left panel*) or GABA_B receptors in the ORNs (*right panel*). The behavioral responses to individual odors are strongly impaired due to this vast manipulation.

REVIEWER #2

(1) Reviewer #2: *Summary: In the wild, fruit flies would usually encounter odors as mixtures of multiple odorant chemicals at differing concentrations and respond to them appropriately. In this manuscript, the authors propose a mechanism for the neuronal integration of conflicting odor stimuli in the second order, antennal lobe projection neurons. Here, the authors identified mixtures of repellent and attractant odors, titrated such that flies were repelled by the mixtures. They then showed that antennal lobe glomeruli responding to the attractant were inhibited when presented along with the repellent. Specific repellent responding glomeruli were shown to laterally inhibit specific attractant responsive glomeruli by using RNAi knock down and optogenetic activation of subsets of repellent glomeruli. This lateral inhibition was shown to rely on different sub-networks of inhibitory, lateral interneurons within the antennal lobe circuit. This segregation of the LN network into different groups allowing different aversive glomeruli to inhibit attractive ones and not the reverse is a significant result. This is a very well-presented manuscript with careful neurophysiological measurements supporting an exciting result. It would benefit greatly from some independent, anatomical verification experiments, which I urge the authors to do.*

(1) RESPONSE: We are happy that this reviewer finds our data significant and exciting. We have thoroughly revised our manuscript according to all suggestions and have tried to clarify all points of criticism.

(2) Reviewer #2: *Major comments: 1. Odor stimulus needs clearer characterization: The main result described here depends on the relative concentrations of the attractive odorant in the mix(+) and mix(-) stimuli. The authors have done a very thorough job of characterizing their odor stimuli by doing both, PID and MS measurements. But the current figure supplement 1a does not capture the relative (un-saturated) PID signal amplitudes for ethyl acetate at the two concentrations it was used. This difference in odor vapor concentrations delivered should be measured and depicted, especially because the authors used oil-dilution rather than air-dilution odor-delivery systems. This data probably already exists and just needs to be added to the plot, so this comment should be easy to address.*

(2) RESPONSE: We have added PID measurements for ethyl acetate and benzaldehyde at the different concentrations used (i.e. ethyl acetate: 10^{-2} , 10^{-3} , 10^{-4} ; benzaldehyde: 10^{-1} , 10^{-2}) to **Supplementary Figure 1**.

(3) Reviewer #2: *2. LN sub-population anatomy not characterized for this result: The central, surprising claim in this study is that different LN sub-networks mediate selective, lateral inhibition between sets of glomeruli. There is some anatomical evidence of this in an EM study in larvae, (Berck, ... Cardona, 2016, eLife 2016;5:e14859) and the authors should definitely discuss their findings in light of this. But more importantly, the authors need to provide an independent anatomical basis to their claims. Is it possible to visualize the anatomy of an LN receiving input from an aversive glomerulus and giving output to a subset of the appetitive ones, consistent with the physiology? One possibility is to double label glomeruli and a subpopulation of LNs (as in figure 7) and show that the only the appropriate aversive glomeruli receive dendrites from the labelled LNs. Also show that only the appropriate appetitive glomeruli receive axons.*

(3) RESPONSE: We thank the reviewer for mentioning the work by Berck et al. (2016) which is highly relevant for our study and we apologize for not citing it in our previous version. We have included that GABAergic LNs have been described in the larval antennal lobe that possess a clear polarity and contribute to postsynaptic inhibition for most glomeruli, while they receive inputs from only a small glomerular subset (lines 518-522). In addition, we agree with the reviewer that we need to provide anatomical evidence that such a polarity also exists for GABAergic LNs in the adult antennal lobe. We have therefore performed additional experiments. First, we employed photoactivatable GFP to label single LNs of the LN populations that are involved in the mixture inhibition (i.e. *NP30-56-Gal4*, *HB4-93-Gal4*) to confirm the existence of individual LNs that connect the *repellent-* and *attractant-responsive* glomeruli (new **Figure 8 e,f**). Second, we followed the reviewer's suggestion to provide independent data for the polarity of LNs. We therefore labeled selectively pre- and postsynapses in the GABAergic LNs of interest (i.e. *NP30-56-Gal4*, *HB4-93-Gal4*) and quantified their densities in the appropriate *repellent-* and *attractant-responsive* glomeruli. These data are included in **Figure 8 g,h** and **Supplementary Figure 9**. Our new data supports our prediction and reveals that GABAergic LNs labeled by *NP3056-Gal4* possess a significantly higher density of postsynapses in the *aversive* glomerulus DL5, while the *attractant-responsive* glomerulus DM3 has more presynapses. We show the same data for the line *HB4-93-Gal4* for glomeruli DL1 versus DM1 and DM4 (lines 396-413 and 513-517)

(4) Reviewer #2: *Minor comments: 1. The authors identify concentrations of attractive odorants and then reduce these to generate the mix(-) stimuli which cause the switch in behavioral response. Did the authors try rather to increase the concentration of the aversive odor to generate the switching mixture?*

(4) RESPONSE: We agree that this would have been interesting to test. However, given that the aversive odors are applied at comparatively high odor concentrations in our study, we did not want to increase their concentrations further. Higher concentrations of benzaldehyde or geosmin would be very difficult to remove and would stick to the odor delivery system and therefore would increase the probability of contaminations.

(5) Reviewer #2: *2. At some points in the manuscript (eg. Line 72) the authors claim to monitor multiple neuronal processing levels, but this is not the case. The model proposed involves glomerular interactions and while some manipulations involved other levels of the circuit, the level monitored and relevant to the message is that of the glomeruli. I suggest re-focusing the language.*

(5) RESPONSE: We have rephrased this sentence (lines 72-73).

REVIEWER #3

(1) Reviewer #3: *An interesting, important, and vexing question is how valences of sensory stimuli are determined and encoded by sensory neurons. In “Odor mixtures of opposing valence unveil inter-glomerular crosstalk in the Drosophila antennal lobe,” authors Mohamed et al. use behavior assays, calcium imaging, optogenetics and other genetic manipulations to test ideas about how attraction and aversion are mediated by defined neural circuits. The manuscript benefits from clear writing, well-designed figures, and abundant data. Unfortunately, the storyline holding all of this together appears to be fundamentally incorrect.*

Main concern

The authors identified one set of antennal lobe glomeruli that can be activated by a chemical (ethyl acetate) that causes behavioral attraction, and another set of glomeruli that can be activated by a different chemical (benzaldehyde) that causes behavioral aversion. The authors then varied the valences of their odor presentation by making mixtures of these chemicals that, depending upon the mixture ratio, would cause either attraction or repulsion. The authors then determined how various glomeruli respond as the mixture ratio was varied. The authors interpret their results to conclude that a specific population of GABAergic local neurons mediates interactions between an attraction channel and an aversion channel within the Drosophila brain. The problem is that there is no evidence that attraction and aversion channels really exist. The authors confirm, as previously known, that at tested concentrations, the aversive odorant, benzaldehyde, most strongly activates glomeruli DL1 and DL5 (lines 122ff). “For simplicity (line 124)”, the authors call these glomeruli “repellent-responsive.” Another set of glomeruli is called “attractant-responsive.” At first this nomenclature seems like a reasonable shorthand as it accurately describes the way these glomeruli respond to the very small set of odors tested here. But, as the manuscript progresses, the authors inappropriately begin to take these terms more seriously, for example by repeatedly referring to “aversive neuronal circuitry” and “attractive neuronal circuitry (i.e. lines 188-190),” and suggesting that specific interneurons modify the aversive and attractive actions of generally opposed circuits that mediate behaviors. This description is internally consistent given the responses to the two or three odorants tested by the authors, but falls apart when considering published data not presented here: the “aversive glomeruli” DL1 and DL5 actually respond best to extremely attractive food and oviposition odors that the authors did not test (such as ethyl benzoate and hexenal, respectively; Laissue and Vosshall, 2008; Lin et al, 2015; etc). Further, odor valence is well-known to be strongly modulated by a variety of factors not tested here, such as satiety level and prior conditioning. Except for the special case of geosmin, the authors present no evidence that general attraction and aversion circuits exist in this part of the brain. Because of this, the authors need to remove this misleading overarching framework from their manuscript. Absent this general interest framework, what remains is not especially novel or compelling: it is evidence of a specific GABAergic connection between specific glomeruli. The function of this connectivity is known only for the role it plays when triggered by a specific odor; other untested odors, as noted above, would lead to very different results and a very different functional interpretation. The existence of this type of connectivity has long been known (Wilson et al, 2003, for example, described important aspects of interglomerular crosstalk in Drosophila; many other authors working in Drosophila and other insects have since described a number of functions served by this crosstalk to transform olfactory representations). Once appropriately rewritten, the results presented here will interest

specialists in Drosophila olfaction who would like to know about the specific functional connectivity between the known glomeruli revealed here, but will frustrate and mislead readers looking for the broader results suggested by the title, abstract, and discussion of this work.

(1) RESPONSE: We must say that we respectfully disagree due to the following reasons: First, the reviewer raised the argument that “*there is no evidence that attraction and aversion channels really exist*”. We are not sure whether the reviewer is aware of the fact that several studies have provided highly convincing evidence for the existence of input channels that are dedicated to solely attractive or aversive olfactory stimuli. Dedicated aversive channels (and circuits) have been shown for the odor geosmin (odor of toxic mold, activating Or56a, targeting DA2 (Stensmyr et al., *Cell*, 2012)), iridomyrmecin, the pheromone of a parasitoid wasp, activating Or49a, targeting DL4 (Ebrahim et al., *PLoS Biol*, 2015), carbon dioxide (activating Gr21a/Gr63a, targeting V (Suh et al., *Nature*, 2004)), and acids (activating Ir64a, targeting DC4 (Ai et al., *Nature*, 2010)). Analogous to the aversive olfactory pathways, also channels that code for attraction have been identified in recent years. For example, olfactory sensory neurons expressing Ir92a (targeting VM1) are highly specific to ammonia and amines, which act as potent attractants in flies (Min et al., *PNAS*, 2013), while ORNs expressing Or83c (targeting DC3) mediate attraction behavior to farnesol (Ronderos et al., *J. Neurosci.*, 2014). Semmelhack & Wang. (*Nature*, 2009) provided evidence that activation of glomeruli DM1 and VA2 mediates attraction towards apple cider vinegar. Furthermore, Bell et al. (*Neuron*, 2016) revealed that optogenetic activation of single ORN types (e.g. DM1, DM2, VA2) induces attraction behavior.

The existence of separate channels for attractive and aversive inputs is further substantiated by tracking the neural circuitry from the periphery to higher processing centers: the axons of PNs that innervate glomeruli responding to attractive odors are topographically segregated from those PNs that are activated by aversive odors (Ebrahim et al., *PLoS Biol*, 2015; Huoviala et al., *BioRxiv*, 2018; Min et al., *PNAS*, 2013; Seki et al., *BMC Biol*, 2017). This suggests that sensory stimuli of opposing valence are represented in spatially distinct neuroanatomic loci within the lateral horn and strongly supports the concept of dedicated circuits for attractive and aversive odors.

Second, the reviewer argues that “*the aversive glomeruli DL1 and DL5 actually respond best to extremely attractive food and oviposition odors that the authors did not test (such as ethyl benzoate and hexenal, respectively)*”. We agree with the reviewer that glomeruli DL1 and DL5 respond also to ethyl benzoate and E2-hexenal. However, the valence of these two odors is somehow contradictory in the literature. MacWilliam et al. (*Neuron*, 2018) show that ethyl benzoate and E2-hexenal induce aversive behavior. In line with this finding is the study of Gao et al. (*PLoS ONE*, 2015) that reveals that E2-hexenal is strongly repellent, while Lin et al. (*eLife*, 2015) shows that E2-hexenal is attractive. Hence these odors cannot be clearly assigned to a certain valence category in contrast to odors (such as benzaldehyde, CO₂, geosmin or vinegar) that induce a robust aversive or attractive response, respectively, in a range of behavioral assays (Seki et al., *BMC Biol*, 2017). What actually matters even more is the behavioral output that is consequently elicited when these glomeruli become activated. For example, ORNs that respond to CO₂ (Gr21a/Gr63a, targeting V glomerulus) are also activated by ethyl benzoate and E2-hexenal (MacWilliam et al., *Neuron*, 2018). However, the CO₂ circuit has been clearly demonstrated to mediate behavioral aversion (Suh et al., *Nature*, 2004; Suh et al., *CurrBiol*, 2007). Like the V glomerulus, DL1 and DL5 also

represent aversive channels due to the following reasons: 1. Knaden et al. (*Cell Rep*, 2012) showed that glomeruli DL1 and DL5 were significantly more activated by the six most aversive odors of a panel of 110 different odors (including all odors from Hallem and Carlson) than by the six most attractive ones, which substantiates the function of glomeruli DL1 and DL5 as “aversive-specific” glomeruli. 2. The axonal projections of DL5 in the lateral horn clearly overlap with other aversive-specific PNs, such as those from DA2 (geosmin) and V (CO₂) and DL4 (iridomyrmecin), and have very similar axonal arborizations in the ventral-posterior lateral horn (Huoviala et al., *BioRxiv*, 2018). It is defined as an aversive PN in the aforementioned study. Notably, PNs innervating DL1 also innervate the ventral-posterior area that is targeted by aversive PNs (Jefferis et al., *Cell*, 2007). 3. Although Or7a responds to the oviposition cue 9-tricosene (Lin et al., *eLife*, 2015), olfactogenetic activation of DL5 (i.e. Or7a) leads to negative oviposition (Chin et al., *Cell Rep*, 2018). 4. By performing additional experiments, we provide evidence that optogenetic activation of glomeruli DL1 and DL5 leads to aversive behavior in comparison to the control flies (new data in **Figure 5g**). Furthermore, we replaced the aversive odor in the binary mixture by optogenetic activation of glomeruli DL1 and DL5 and could mimic the behavioral response to *MIX(+)* and *MIX(-)* (**Figure 5g**). Hence, we provide new and solid evidence that glomeruli DL1 and DL5 represent aversive input channels. To meet the concerns of the reviewer, we have now added the above mentioned arguments to the manuscript to strengthen the notion that activation of DL1 and DL5 mediate aversion. Regarding the attractant-responsive glomeruli, we have performed additional experiment as proposed by the reviewer and silenced DM1/DM4 to show that attraction towards the attractive odor ethyl acetate at both concentrations (i.e. 10⁻² and 10⁻³) and to both mixtures is abolished (**Suppl. Fig. 4c**; see further details below at point (8)).

Third, the reviewer raises the concern that the existence of a specific GABAergic connection between specific glomeruli has long been known. We agree that several papers have already provided evidence for the existence of GABAergic lateral inhibition in the antennal lobe in various insect species over the last decades. We cite and discuss these studies in our introduction and discussion. However, in this study we report several novel discoveries that extend the previous work: (1) we provide evidence for a selective glomerular crosstalk, (2) we identify the glomeruli involved, (3) we show that the crosstalk takes place between glomeruli that respond mainly to odors with opposing valence, and, (4) most importantly, we link this crosstalk to the behavioral output. In summary we show for the first time the presence and function of glomeruli-specific lateral inhibition in mediating the fine balance between odor-induced attraction and aversion.

(2) Reviewer #3: *Specific concerns: (1) line 20: “attractant-responsive input channels”; line 25: “attractant- and repellent-specific circuits”, line 228-9: “a specific ORN population that encodes aversive odors”, etc, throughout the manuscript: as outlined above, descriptions like these are false and highly misleading.*

(2) RESPONSE: As explained in detail above, we are convinced that attractive and aversive input channels exist. However, in order to meet the reviewer’s concern, we have rephrased some of our terms regarding the different channels and replaced ‘specific’ by ‘responsive’.

(3) Reviewer #3: *(2) lines 105-6: “Ethyl acetate and benzaldehyde evoke activity in, mostly, non-overlapping glomeruli.” This is true, but they are clearly not separate channels; both odors evoke activity in multiple glomeruli getting input from receptor neurons expressing*

Or67a, Or85b, Or98a, Or19a (Hallam and Carlson, fig. 6a), so there is some overlap in evoked activity.

(3) RESPONSE: We agree that the odors benzaldehyde and ethyl acetate evoke activity in multiple glomeruli. However, contrary to the statement of the reviewer, the SSR data by Hallam and Carlson (Cell, 2006) do not show any overlap, while the DoOR database (Münch et al., *Sci. Reports*, 2016) shows one case of an overlap for Or85e (targeting glomerulus VC1). Hence, our statement that “ethyl acetate and benzaldehyde evoke activity in, mostly, non-overlapping glomeruli” is completely correct. In addition, we do not state that these two odors are coded by completely separate input channels.

(4) Reviewer #3: (3) lines 124-5: “For simplicity, we name this subset of glomeruli henceforth attractant-responsive or repellent-responsive glomeruli, respectively.” This is misleading for two reasons. First, as noted above, the “repellent-responsive glomeruli” also respond very well to attractive odors not discussed in this manuscript. Second, other glomeruli also respond to the test odors (Sup. Fig. 2), so these subsets are incomplete.

(4) RESPONSE: As explicated in detail above, we provide clear evidence that the *repellent-responsive* glomeruli mediate aversion and can be regarded as aversive input channels. We fully agree that also other glomeruli respond to the attractive and repellent odors and therefore included 34 glomeruli into our analysis. We present the data of those glomeruli that are responding strongest (i.e. 6 glomeruli) in the main figures, while we show the data of the remaining 28 glomeruli in the Supplementary figures (**Suppl. Fig. 2 and 3**). Notably, we observed mixture interactions in form of inhibition only in four out of these 34 glomeruli (lines 131-134) and therefore focused our further experiments on these glomeruli to elucidate the underlying neuronal mechanism causing this mixture inhibition. We clarified this point now stronger in our result section.

(5) Reviewer #3: (4) line 129: “we noticed strong inhibition in four out of the 34 glomeruli” This result is impossible to interpret because, as the authors themselves correctly note later in the manuscript: “inhibitory responses are difficult to capture with calcium imaging (line 471-2).” It is impossible to say whether other glomeruli also receive inhibitory input.

(5) RESPONSE: We would like to clarify this point which is based on a misunderstanding. When we stated that it is difficult to capture inhibitory responses with calcium imaging, we thought of hyperpolarization. However, in the statement that the reviewer mentioned (“we noticed strong inhibition in four out of the 34 glomeruli”), we were describing mixture inhibition, which is well visible with calcium imaging, since the neurons/glomerulus becomes activated by the attractant but becomes significantly less activated by this compound in the presence of the repellent. We have revised these sentences to avoid misunderstandings (lines 132 and 538).

(6) Reviewer #3: (5) line 137: “we next examined whether benzaldehyde always inhibits these four attractant-responsive glomeruli when it reduces behavioral attraction in the mixture.” This is an overstatement of the evidence: given such a limited stimulus set, the authors cannot conclude what always happens.

(6) RESPONSE: We apologize for the misleading wording and have rephrased this sentence to “We next examined whether the mixture inhibition is concentration- or ratio-dependent.” (lines 140-141).

(7) Reviewer #3: (6) line 154-5: “methyl salicylate... which activates only glomerulus DL1”
No, methyl salicylate also activates VC2 (Olsen et al, 2007).

(7) RESPONSE: It is correct that methyl salicylate also activates glomerulus VC2 (Münch et al., Sci Rep., 2016), while Olsen et al. (2007) reports activation of glomerulus VA7I. However, those studies used a 10-fold higher concentration of methyl salicylate. At the concentration used in our study we observe only strong activation of glomerulus DL1 (please see data in **Suppl. Fig. 3**). We rephrased this sentence to “... which activates only glomerulus DL1 at the used concentration of 10^{-3} ”. (lines 158-160).

(8) Reviewer #3: (7) around lines 220: The authors suggest that a decrease in activity in DM1-DM4 results in decreased attraction to attractive odorants, but this was not directly tested. Are OR mutants available in which DM1-DM4 activity (or at least DM1&DM4) is selectively decreased in response to the attractants? To strengthen their argument the authors could test such mutants with the Flywalk or Tmaze assays.

(8) RESPONSE: Thank you for this excellent suggestion. We have performed additional experiments and silenced the input to glomeruli DM1 and DM4 using *UAS-Kir2.1*. and tested those flies and their parental controls in the T-maze assay. Notably, odor attraction towards ethyl acetate at both concentrations (i.e. 10^{-2} and 10^{-3}) and to both mixtures is abolished when the input to DM1 and DM4 has been silenced, while the parental controls resemble the response of the wild type flies. Flies with silenced DM1/DM4 input show even a behavioral aversion towards *MIX(+)* and *MIX(-)*. This new data is now included in **Supplementary Figure 4c** and discussed in the manuscript (lines 228-231).

(9) Reviewer #3: (8) lines 222-3: the authors described the data in figure 4f as “silencing glomerulus DL5 and consequently abolishing the inhibition in glomeruli DM2 and DM3”. But a substantial difference between activity evoked by ETA and *MIX(-)* in DM2 remains – inhibition is reduced but not abolished.

(9) RESPONSE: We have rephrased this sentence to “... silencing glomerulus DL5 and consequently abolishing the inhibition in glomerulus DM3 and reducing in DM2 ...” (lines 234-235).

(10) Reviewer #3: (9) lines 224-5: “this suggests that not all glomeruli are crucial for odor valence coding.” This is a vast overstatement. The authors tested only two odors. It is perfectly possible that silencing DM5 may have effects on valence coding for other odors. Their evidence shows these glomeruli are not crucial for valence coding under the extremely limited set of conditions tested here.

(10) RESPONSE: We have rephrased this sentence to “We therefore hypothesize that not all activated glomeruli might be crucial for odor valence coding - an assumption that needs to be tested in future experiments.” (lines 236-238).

(11) Reviewer #3: (10) lines 312-315 and 442-446: The authors argue that GABA_B receptors act pre-synaptically and GABA_A receptors act post-synaptically in inhibiting the attractant-responsive glomeruli. However, the RNAi experiment that is shown in fig. 6 and sup. fig. 6 used *Gal4* lines that are not specific to the ORNs and PNs of interest, leaving open the possibility that inhibition is mediated by other, unknown neurons. The RNAi

experiment should be repeated using cell-specific Gal4 lines (for example, Or7a or Or10 gal4 for ORNs).

(11) RESPONSE: We are not sure whether the reviewer is aware of the fact that expressing RNAi against GABAergic receptors under control of *Or7a*- or *Or10a*-*GAL4* would not further clarify the inhibitory mixture mechanism, since these ORs-expressing neurons target the *repellent-responsive* glomeruli that do not show any mixture inhibition. Also, if we would express RNAi against GABAergic receptors selectively in ORNs of the *attractant-responsive* glomeruli (i.e. DM1-DM4), we would not elucidate the circuit further and would not be able to draw more specific conclusions than the ones we already state. We have carefully considered the reviewer's comment and thought of other circuit elements that could mediate the inhibition, but could not think of any. Since ORNs are uniglomerular and all ORNs expressing the same OR converge onto one glomerulus, other ORN types (i.e. innervating other glomeruli) cannot account for the effect observed. In addition there are no specific PN lines available that label exclusively PNs from our glomeruli of interest.

Reviewers' Comments:

Reviewer #1:

Remarks to the Author:

I read the response of the authors to my comments and to other reviewers' comments carefully.

I see that all of the reviewer's comments are carefully addressed in the revised manuscript. The additional behavioral, pharmacological and anatomical data further supports the validity of authors arguments in response to the points raised by reviewer 3.

I do not share the view of reviewer 3 about the value of this presented work only for specialists in drosophila olfaction. The presented manuscript shows for the first time that a novel type of rather targeted lateral inhibition can play an important role in comparing sensory inputs received by parallel channels carrying information about hedonically different stimuli. Authors successfully showed that such targeted lateral inhibition is important for animals to make correct decisions, when they encounter odors with different valences. Later, they went through great lengths to parse apart the mechanisms underlying such targeted lateral inhibition. The presented work is novel, extremely interesting and of very high quality. The question for the rest of field of sensory systems neuroscience will now be to investigate whether similar mechanisms also exist in other sensory modalities, as well as other animals. I therefore find the presented findings very exciting and suitable for publication in Nature Communications.

Reviewer #2:

Remarks to the Author:

The authors have adequately responded to the concerns I raised in my initial review, and also those raised by reviewer 1, in my opinion.

The central issue raised by reviewer 3 is to challenge the idea that there are glomeruli devoted mainly to the encoding of appetitive or aversive odors. It is true that most glomeruli have responses that are not valence-specific. However, as discussed by the authors at length in the rebuttal, it is also clear that a small set of glomeruli do tend to respond to odors of specific valence. That addresses input valence specificity. As for output valence specificity, the activation of specific glomeruli (fig. 5g, also see rebuttal) causes behavioral responses as though valence-specific channels were activated. If these special glomeruli have responses that are valence-specific and their activation causes behavior that is valence-specific – I think it is fair to call them valence-specific channels. Perhaps changes made by the authors in the language in the paper to say that only very few, special glomeruli show valence-specificity would avoid confusing readers and address the concerns of reviewer 3.

Reviewer #3:

Remarks to the Author:

With this resubmission authors Mohamed et al. have strengthened some components of their work, "Odor mixtures of opposing valence unveil inter-glomerular crosstalk in the Drosophila antennal lobe," with additional experiments and text revisions. I want to acknowledge that the authors have done a great deal of work, and much of it is of very high quality. However, they have not resolved my main concern, that the storyline holding all of their work together is fundamentally incorrect. Unless resolved, this concern should preclude publication not only in Nature Communications, but anywhere.

The neural circuitry of the insect antennal lobe has been studied for many years by many groups. It has been known for many years that this circuitry includes interacting excitatory and inhibitory neurons that influence each other on multiple time scales. The well-documented functions of these interactions include response normalization, the generation of temporally structured spiking patterns that carry information about odors, the oscillatory synchronization of projection neuron output, and the distribution of olfactory information across multiple glomeruli such that responses of projection generally are more broadly tuned than responses of their presynaptic olfactory receptor neurons. To this list of functions performed by the antennal lobe Mohamed et al. now add the determination of valence.

I can appreciate the appeal of this claim. Valence decisions obviously have to be made somewhere in the brain, and in a few highly specialized pheromone-like cases, where an odorant holds critical meaning to the animal, it does indeed appear that valence decisions are made very early in the olfactory pathway. In my initial review I acknowledged the case of geosmin, an odorant signaling the presence of a toxic mold lethal to fruit flies. In their rebuttal letter, the authors point to a couple more special cases: iridomyrmecin, a pheromone produced by a dangerous parasitoid wasp; and CO₂, which serves as an alarm pheromone in fruit flies. These odorants have been shown to activate relatively few receptor types and glomeruli (although the situation is more complex than the authors let on for iridomyrmecin, whose primary receptor Or49a responds to many additional odorants including attractive fruit odors such as 2-heptanone, 1-hexanol, pentyl acetate, heptanol, even at 10e-2 dilutions (table S1, Kraher et al, 2008; Ibba et al, 2010), and for CO₂, whose processing requires the interaction of multiple excitatory and inhibitory inputs to the V glomerulus).

What the authors are implicitly arguing in the present manuscript is something different, that the antennal lobe contains "aversive" and "attractive" circuitry not just for highly specialized pheromone-like odorants, but for other general odorants that have not been described as holding critical meaning to the animal, such as ethyl acetate and benzaldehyde. The authors claim that the aversive or attractive circuitry receives input from aversive or attractive receptors and projects to specialized aversive or attractive brain areas downstream. Here, the authors are making a fairly radical claim that is difficult to square with well-accepted facts including these: (1) as Hallam and Carlson and many other authors have shown, most odorants activate multiple, and sometimes large numbers of receptors; (2) simple conditioning paradigms make it trivially easy to quickly and dramatically change the valence of odors in flies and other animals; (3) the valence of an odor can depend strongly on the animal's internal state, such as hunger vs. satiety. I raised these concerns in my previous review, but in their rebuttal the authors ignored them.

Even fairly radical claims can be correct. But to be convincing they need to be supported by serious efforts to falsify them. I don't think this is an unfair type of demand or an excessive one in the case of this manuscript – it's basic science. If the authors claim to identify "aversive circuitry" then they really need to convince readers that this circuitry is not activated by attractive odorants. Here, the authors run into trouble because, as they neglected to mention in their manuscript, their "aversive circuitry" is actually most strongly activated by extremely attractive food and oviposition odors that the authors did not test (such as ethyl benzoate and hexenal, respectively). In their rebuttal letter the authors acknowledge that their "aversive" glomeruli DL1 and DL5 respond also to ethyl benzoate and E2-hexenal, but they then argue that these food and oviposition odors may not always be attractive, depending upon the circumstances.

But here the authors are making my argument for me. Except for special cases like pheromones and pheromone-like odors, the answer to the question: "Is this odor attractive or aversive" is usually not yes or no, but rather "it depends upon the circumstances." It is difficult to square this reality with the hard-wired antennal lobe circuit model proposed by the authors.

And, critically, the authors have not made a serious attempt to falsify their overarching model. Instead of arguing in a rebuttal letter that food and oviposition odors may not be attractive, they need to actually test them. The authors tested an extremely small number of odorants, and it is not clear why they selected them from the wide olfactory world. To rigorously attempt to falsify their model, the authors need to draw up a reasonable list of odorants at a reasonable range of concentrations known to be attractive to the fly and test whether they activate the "aversive" circuit. This list should certainly include ethyl benzoate and E2-hexenal. The authors also need to draw up a reasonable list of odorants at a reasonable range of concentrations known to be aversive to the fly and test whether they activate the "attractive" circuit. These lists of odorants should be sufficiently long and broad to convince readers that the authors have made a serious effort to falsify their model. If the results are clear-cut, they will provide convincing evidence for their model. But this evidence does not yet exist, and in its absence the authors' overarching storyline is fundamentally unsupportable.

Response to reviewer comments

Mohamed et al. (NCOMMS-18-21157A)

REVIEWER #1

(1) Reviewer #1: *I read the response of the authors to my comments and to other reviewers' comments carefully.*

I see that all of the reviewer's comments are carefully addressed in the revised manuscript. The additional behavioral, pharmacological and anatomical data further supports the validity of authors arguments in response to the points raised by reviewer 3.

I do not share the view of reviewer 3 about the value of this presented work only for specialists in drosophila olfaction. The presented manuscript shows for the first time that a novel type of rather targeted lateral inhibition can play an important role in comparing sensory inputs received by parallel channels carrying information about hedonically different stimuli. Authors successfully showed that such targeted lateral inhibition is important for animals to make correct decisions, when they encounter odors with different valances. Later, they went through great lengths to parse apart the mechanisms underlying such targeted lateral inhibition. The presented work is novel, extremely interesting and of very high quality. The question for the rest of field of sensory systems neuroscience will now be to investigate whether similar mechanisms also exist in other sensory modalities, as well as other animals. I therefore find the presented findings very exciting and suitable for publication in Nature Communications.

(1) RESPONSE: Thank you very much for your overall positive feedback regarding our revised manuscript!

REVIEWER #2

(1) Reviewer #2: *The authors have adequately responded to the concerns I raised in my initial review, and also those raised by reviewer 1, in my opinion.*

The central issue raised by reviewer 3 is to challenge the idea that there are glomeruli devoted mainly to the encoding of appetitive or aversive odors. It is true that most glomeruli have responses that are not valence-specific. However, as discussed by the authors at length in the rebuttal, it is also clear that a small set of glomeruli do tend to respond to odors of specific valence. That addresses input valence specificity. As for output valence specificity, the activation of specific glomeruli (fig. 5g, also see rebuttal) causes behavioral responses as though valence-specific channels were activated. If these special glomeruli have responses that are valence-specific and their activation causes behavior that is valence-specific – I think it is fair to call them valence-specific channels. Perhaps changes made by the authors in the language in the paper to say that only very few, special glomeruli show valence-specificity would avoid confusing readers and address the concerns of reviewer 3.

(1) RESPONSE: Thank you for your feedback to our revision and the point that you have raised! As suggested, we have made appropriate textual adjustments to stress the point that only very few, special glomeruli show valence-specificity (pages 5, 9, 10, 16).

REVIEWER #3

(1) Reviewer #3: *With this resubmission authors Mohamed et al. have strengthened some components of their work, “Odor mixtures of opposing valence unveil inter-glomerular crosstalk in the Drosophila antennal lobe,” with additional experiments and text revisions. I want to acknowledge that the authors have done a great deal of work, and much of it is of very high quality. However, they have not resolved my main concern, that the storyline holding all of their work together is fundamentally incorrect. Unless resolved, this concern should preclude publication not only in Nature Communications, but anywhere.*

(1) RESPONSE: *We are pleased that the reviewer acknowledges the high quality of our work, our text modifications and additional experiments in the revised version of our manuscript. However, we still strongly disagree with the reviewer’s major concern that the storyline of our study is incorrect as justified in detail below!*

(2) Reviewer #3: *The neural circuitry of the insect antennal lobe has been studied for many years by many groups. It has been known for many years that this circuitry includes interacting excitatory and inhibitory neurons that influence each other on multiple time scales. The well-documented functions of these interactions include response normalization, the generation of temporally structured spiking patterns that carry information about odors, the oscillatory synchronization of projection neuron output, and the distribution of olfactory information across multiple glomeruli such that responses of projection generally are more broadly tuned than responses of their presynaptic olfactory receptor neurons. To this list of functions performed by the antennal lobe Mohamed et al. now add the determination of valence.*

I can appreciate the appeal of this claim. Valence decisions obviously have to be made somewhere in the brain, and in a few highly specialized pheromone-like cases, where an odorant holds critical meaning to the animal, it does indeed appear that valence decisions are made very early in the olfactory pathway. In my initial review I acknowledged the case of geosmin, an odorant signaling the presence of a toxic mold lethal to fruit flies. In their rebuttal letter, the authors point to a couple more special cases: iridomyrmecin, a pheromone produced by a dangerous parasitoid wasp; and CO₂, which serves as an alarm pheromone in fruit flies. These odorants have been shown to activate relatively few receptor types and glomeruli (although the situation is more complex than the authors let on for iridomyrmecin, whose primary receptor Or49a responds to many additional odorants including attractive fruit odors such as 2-heptanone, 1-hexanol, pentyl acetate, heptanol, even at 10e-2 dilutions (table S1, Kraher et al, 2008; Ibbá et al, 2010), and for CO₂, whose processing requires the interaction of multiple excitatory and inhibitory inputs to the V glomerulus).

What the authors are implicitly arguing in the present manuscript is something different, that the antennal lobe contains “aversive” and “attractive” circuitry not just for highly specialized pheromone-like odorants, but for other general odorants that have not been described as holding critical meaning to the animal, such as ethyl acetate and benzaldehyde. The authors claim that the aversive or attractive circuitry receives input from aversive or attractive receptors and projects to specialized aversive or attractive brain areas downstream. Here, the authors are making a fairly radical claim that is difficult to square with well-accepted facts including these: (1) as Hallam and Carlson and many other authors have shown, most odorants activate multiple, and sometimes large numbers of receptors; (2) simple

conditioning paradigms make it trivially easy to quickly and dramatically change the valence of odors in flies and other animals; (3) the valence of an odor can depend strongly on the animal's internal state, such as hunger vs. satiety. I raised these concerns in my previous review, but in their rebuttal the authors ignored them.

Even fairly radical claims can be correct. But to be convincing they need to be supported by serious efforts to falsify them. I don't think this is an unfair type of demand or an excessive one in the case of this manuscript – it's basic science. If the authors claim to identify "aversive circuitry" then they really need to convince readers that this circuitry is not activated by attractive odorants. Here, the authors run into trouble because, as they neglected to mention in their manuscript, their "aversive circuitry" is actually most strongly activated by extremely attractive food and oviposition odors that the authors did not test (such as ethyl benzoate and hexenal, respectively). In their rebuttal letter the authors acknowledge that their "aversive" glomeruli DL1 and DL5 respond also to ethyl benzoate and E2-hexenal, but they then argue that these food and oviposition odors may not always be attractive, depending upon the circumstances.

But here the authors are making my argument for me. Except for special cases like pheromones and pheromone-like odors, the answer to the question: "Is this odor attractive or aversive" is usually not yes or no, but rather "it depends upon the circumstances." It is difficult to square this reality with the hard-wired antennal lobe circuit model proposed by the authors.

(2) RESPONSE: The reviewer continues his/her concerns about whether "aversive" and "attractive" circuits really code for aversion and attraction, respectively, and ignores our newly added crucial experiments which demonstrate that artificial activation of glomeruli DL1 and/or DL5 (i.e. the *repellent-responsive glomeruli*) induced behavioral aversion (Figure 5g). In addition, silencing glomeruli DM1 and DM4 (i.e. two of the *attractant-responsive glomeruli*) abolished attraction behavior (Figure S4c). We are convinced (in agreement with Reviewer 1 and 2) that these experiments provide crucial and solid evidence that these channels can be defined as valence-specific channels!

The reviewer argues that an aversive or attractive circuit should not be activated by attractive or aversive odors, respectively. However, that view is incorrect and not substantiated by the literature. An attractive odorant may indeed activate some aversive channels beside their main activation of the attractive circuitry (or the other way around). However, what actually matters is the behavioral output that an odorant or activation of a specific glomerulus is eliciting. For example, CO₂ which represents a highly aversive stimulus to the fly, is encoded by Gr21a/Gr63a. Notably, these CO₂ receptors are also activated by other attractive odors (including those food odors that the reviewer mentioned) (MacWilliam et al., 2018, Neuron). However, this circuit is one of the best-known aversive circuits, since it has been demonstrated that artificial activation of neurons expressing Gr21a/Gr63a leads to strong behavioral aversion (Suh et al., 2007, Curr Biol).

In case of the odor iridomyrmecin, the reviewer contradicted him/herself, in which s/he agrees that the iridomyrmecin circuit represents an aversive channel (as demonstrated by optogenetic activation (Ebrahim et al., 2015, PLoS Biol)), although its receptor Or49a is also slightly activated by attractive fruit odors such as 2-heptanone, 1-hexanol, pentyl acetate, and heptanol. However, since the attractive odors activate several other glomeruli, the combinatorial code in the end determines the behavioral output and not just the activation of the aversive channel.

In our study we intentionally selected odors with a clear and robust behavioral output which is independent of the behavioral assay used. We agree that the internal state of the fly matters, as this has been shown in several studies. We therefore kept the internal state of the flies constant and performed our experiments always with female flies (4-6 days after eclosion) that have been starved for 24h (as described in the Methods). We mention now in the Results section that the internal state has been shown to influence odor-guided behavior as well as odor-evoked responses in the AL and therefore needs to be kept constant (page 5, lines 106-109).

(3) Reviewer #3: *And, critically, the authors have not made a serious attempt to falsify their overarching model. Instead of arguing in a rebuttal letter that food and oviposition odors may not be attractive, they need to actually test them. The authors tested an extremely small number of odorants, and it is not clear why they selected them from the wide olfactory world. To rigorously attempt to falsify their model, the authors need to draw up a reasonable list of odorants at a reasonable range of concentrations known to be attractive to the fly and test whether they activate the “aversive” circuit. This list should certainly include ethyl benzoate and E2-hexenal. The authors also need to draw up a reasonable list of odorants at a reasonable range of concentrations known to be aversive to the fly and test whether they activate the “attractive” circuit. These lists of odorants should be sufficiently long and broad to convince readers that the authors have made a serious effort to falsify their model. If the results are clear-cut, they will provide convincing evidence for their model. But this evidence does not yet exist, and in its absence the authors’ overarching storyline is fundamentally unsupportable.*

(3) RESPONSE: Our results that certain input channels can code for different hedonic valences (as we demonstrate in Figure 5g and Figure S4c) are reminiscent with previous studies showing that artificial activation or silencing of individual olfactory channels can lead to either behavioral attraction or aversion (e.g. Bell and Wilson, 2016, Neuron; Chin et al., 2018, Cell Rep.; Dweck et al., 2013, Curr Biol). The suggested experiments of the reviewer to test our glomeruli of interest with a “reasonable list of odorants at a reasonable range of concentrations that are attractive or aversive to the fly” will not yield to any meaningful outcome for our study and would not further substantiate our inter-glomerular crosstalk. These experiments would only describe the response profiles of a subgroup of glomeruli which is out of the scope of our study. In addition, several studies over the last years have already characterized the odor profiles of many glomeruli, including ours, and are available in the DoOR database (Münch et al., 2016, Sci Rep).

In conclusion let us say that we have done the most crucial experiments using optogenetic activation and silencing and are convinced (in agreement with Reviewer 1 and 2) that these experiments provide convincing and solid evidence that these channels can be defined as valence-specific channels. In order to address the concerns of Reviewer 3 we have stressed the point that only very few, special glomeruli show valence-specificity (pages 5, 9, 10, 16).

In addition, we have added a paragraph in the discussion section that reflects the alternate view expressed by Reviewer 3 regarding the activation of the aversive circuit by non-aversive and attractive odors (page 16): *“It is important to mention that our subset of repellent-responsive glomeruli does also respond to non-aversive and even partly attractive odors, such as E2-hexenal and ethyl benzoate (Hallem and Carlson, Cell, 2006). However, an attractive odorant may indeed activate some aversive input channels beside their main activation of the attractive circuitry (or the other way around). What actually matters is the*

behavioral output that is consequently elicited when a specific glomerulus becomes activated. For example, ORNs that respond to CO₂ are also activated by ethyl benzoate and E2-hexenal (MacWilliam et al., Neuron, 2018). However, the CO₂ circuit has been clearly demonstrated to mediate behavioral aversion (Suh et al., Nature, 2004; Suh et al., CurrBiol, 2007). Following this argument, artificial activation of glomeruli DL1 and/or DL5 as shown in this study leads to aversive behavior, while silencing the attractant-responsive glomeruli DM1 and DM4 abolished attraction to the attractant. These experiments provide evidence that activation of the repellent- and attractant-responsive glomeruli causes a valence-specific behavior, and can therefore be defined as attractive or aversive input channels, respectively.”